# From Memory to Reasoning: Generative Models Enable Explainable Continual Learning

## Abstract

Class Incremental Learning (CIL) aims to enable models to acquire new knowledge over time without forgetting previous tasks. However, existing CIL approaches primarily rely on discriminative modeling, making them prone to catastrophic forgetting due to parameter expansion and lacking transparency in the prediction process, which limits their reliability in real-world settings. In contrast, human continuous learning is inherently generative and interpretable. Humans integrate new concepts by focusing on salient visual details and linking them to prior semantic structures, forming traceable chains of reasoning. Therefore, we propose the Generative Explainable Class-Incremental Learning (GECL), a pioneering generative and interpretable CIL framework designed to address these challenges. GECL employs soft-label-guided visual augmentation to focus attention on the most discriminative image regions and utilizes large language models (LLMs) to construct structured semantic attributes. This approach eliminates the need for expanding classification heads, preventing parameter overwriting and preserving previous knowledge. These semantic attributes achieve fine-grained alignment with visual features through entropy-regularized optimal distribution matching, where a cost matrix explicitly quantifies each attribute-region contribution, generating transparent attribute-region reasoning chains. Experiments across natural scenes and fine-grained datasets demonstrate that GECL balances high accuracy, low forgetting rates, and interpretability, marking a promising step toward safe and reliable continuous learning.

## 1 Introduction

In dynamic, open real-world environments, models often require the ability to continuously absorb emerging knowledge about new classes to adapt to environmental changes and task evolution. Class Incremental learning (CIL,Hayes et al. (2020),Shin et al. (2017),Sun et al. (2023)) is a key learning paradigm designed for this purpose, enabling models to integrate new knowledge without access to prior knowledge. However, despite significant performance improvements in traditional discriminative CIL methods(Thengane et al. (2022b),Wang et al. (2022e),Smith et al. (2023b)), they still face two fundamental challenges. First, the incremental training process can lead to catastrophic forgetting of old knowledge. Second, the decisions of models lack interpretability. The former limits the system's ability to sustain knowledge accumulation, while the latter hinders its deployment in real-world scenarios. In high-stakes scenarios such as medical diagnosis, autonomous driving, and industrial inspection, model's predictions must not only be accurate but also provide transparent decision-making processes. This enables users to understand the basis for judgments, identify potential errors, and implement effective oversight. Existing CL methods(Douillard et al. (2022b),Wang et al. (2022c)) typically rely on parameter overwriting and feature rewriting in classification heads to absorb new classes. This gradually erases traces of old knowledge while new knowledge lacks explicit semantic anchors, ultimately turning the model into a "black box" devoid of traceable reasoning. This non-explanatory knowledge accumulation severely limits CL's applicability in real-world open environments, making it difficult to support long-term autonomous learning for practical applications.

In recent years, with the rise of Large Language Models (LLMs), researchers have begun exploring their integration into CIL to mitigate the challenges of learning new classes by leveraging their rich world knowledge and cross-modal representation capabilities. For instance,

Cao et al. (2024)proposed a generative incremental learning framework using LLMs. This approach constructs template-based textual descriptions for each class (e.g., "this is a [class]") and combines them with a pre-trained visual-text encoder to achieve class extension. Such methods significantly reduce reliance on large annotated datasets and demonstrate potential in tasks like fine-grained recognition, natural scene understanding, and remote sensing analysis.

However, LLMs-driven CIL approaches face three challenges in advancing toward a generalizable paradigm. First, their prediction processes lack interpretability—models cannot provide traceable decision criteria, and users cannot understand their reasoning or analyze error sources. Second, they rely on single-sentence, template-based category descriptions, lacking structured semantic knowledge such as attributes and relationships. This limits their ability to model inter-class connections and facilitate knowledge transfer. Existing research(Sarfraz et al., 2024) indicates that explicitly modeling semantic structures is crucial for efficient knowledge integration. Third, while they avoid fine-tuning visual encoders, they lack explicit constraints on the discrimina-

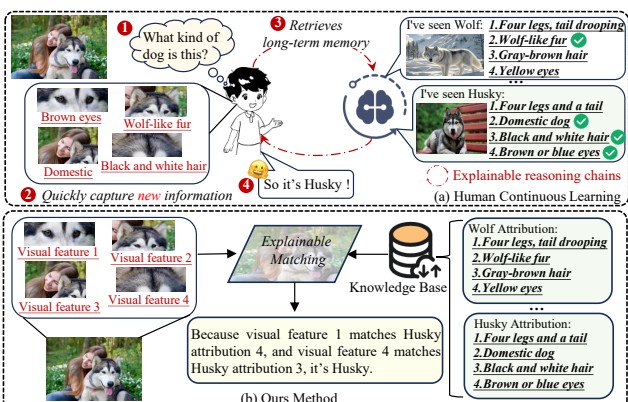

Figure 1: Human-inspired explainable incremental learning. Our method mimics human reasoning by matching new visual features with stored semantic attributions, forming explainable reasoning chains for generative and interpretable incremental learning.

tive boundaries of old classes. This leads to accumulated classification biases and knowledge forgetting when new classes are introduced—an issue also highlighted in studies like Federated Class-Incremental Learning. These shortcomings indicate that neither discriminative approaches nor generative methods based on LLMs—despite improving classification accuracy—have fully addressed the more fundamental scientific question: "How can models provide interpretable reasoning chains while continuously learning?"

To this end, inspired by the human learning paradigm, we propose a Generative Explainable Class-Incremental Learning (GECL) framework. As shown in Fig. 1, aiming to endow CIL systems with intrinsic explainability and structured reasoning capabilities. Compared to previous work such as GMM (Cao et al., 2024), GECL introduces structured attribute generation instead of template-based class statements. The core idea of GECL is to construct a unified visual-semantic feature space through generative semantic modeling and distribution matching while absorbing new class knowledge. It explicitly reconstructs the classification process as a distributional matching inference between visual regions and semantic attributes, thereby achieving continuous knowledge expansion while maintaining transparent decision-making. GECL is not intended as a universal continual learning solution. Rather, it aims to address class-incremental image classification where the prediction must be accompanied by transparent, human-understandable visual reasoning.The main contributions of our approach include:

1.We propose a Generative Explainable Class-Incremental Learning (GECL) framework, introducing knowledge bases and explainable reasoning into the CL scenario to simultaneously address knowledge expansion without forgetting and transparent, traceable decision-making.

2.We design a visual enhancement module to focus on visual patterns highly correlated with classification. Through the matching module, we refine the association between visual attributes and structured semantic features, enabling human-like, interpretable, and traceable reasoning chains.

3.Experiments across three widely-used datasets demonstrate that GECL not only efficiently integrates new classes but also generates traceable reasoning grounds. It significantly outperforms existing baseline in classification accuracy, interpretability quality, and forgetting resistance.

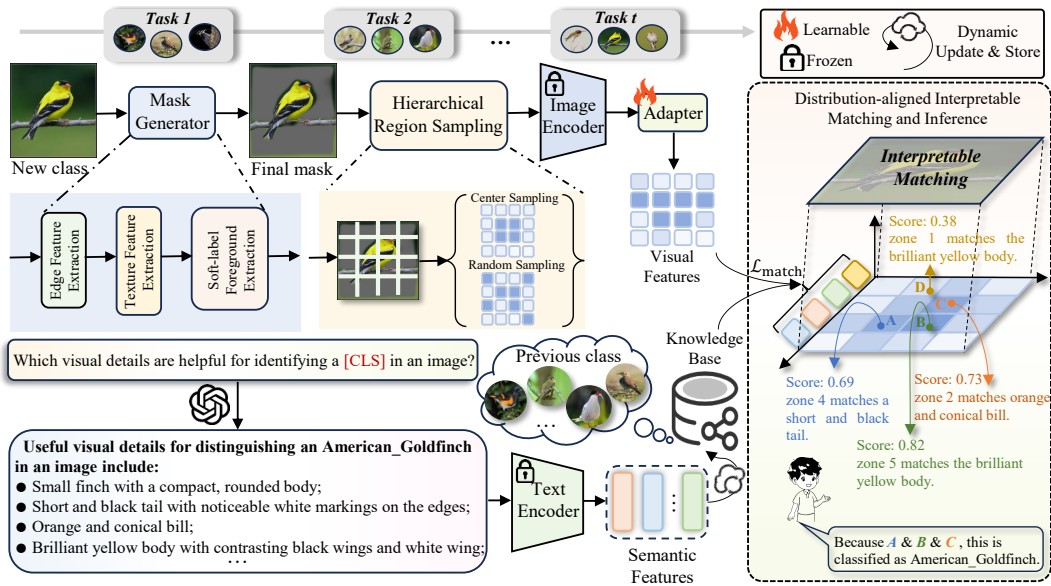

Figure 2: Framework of the GECL. We enhance visual features with soft labels and performs local feature sampling to obtain visual features. By integrating semantic attributes generated by LLMs, we align these features with explainable visual-language matching, yielding a transparent reasoning chain and interpretable outputs, enhancing the credibility of the model's predictions.

## 2 PRELIMINARY

We consider the standard incremental learning setup with a sequence of tasks $t = 1, ..., T$. Task $T$ introduces a disjoint set of new classes $\mathcal{Y}_t$; denote by $\mathcal{Y}^{(t)} = \bigcup_{i=1}^{t} \mathcal{Y}_i$ the set of all classes observed up to task $t$. Let $\mathcal{D}_t$ be the data distribution of task $t$ over image-label pairs $(x, y)$ with $y \in \mathcal{Y}_t$. The classical learning objective is to learn parameters $\theta$ for a model $f_\theta$ that minimize the accumulated risk across tasks:

$$\min_\theta \sum_{t=1}^{T} \mathbb{E}_{(x,y) \sim \mathcal{D}_t} \left[ \mathcal{L}(f_\theta(x), y) \right]. \tag{1}$$

Typical discriminative remedies attempt to approximate Eq. 1 but tend to be opaque and remain vulnerable to catastrophic forgetting due to parameter overwriting. We instead re-formulate the classification decision as matching visual region distributions to semantic attribute distributions in a shared embedding space. Replacing the per-sample discriminative loss in Eq. 1 by this matching cost yields our core optimization objective:

$$\min_\theta \sum_{t=1}^{T} \mathbb{E}_{(x,y) \sim \mathcal{D}_t} \left[ \mathrm{DM}_\lambda \left( V_\theta(x), S(y) \right) \right] + \mathcal{R}(\theta), \tag{2}$$

where $\mathrm{DM}_\lambda(V, S)$ denoted the distribution matching of visual features $V$ and semantic features $S$, and $\mathcal{R}(\theta)$ is a lightweight regularizer. Below we define $\mathrm{DM}_\lambda(V, S)$ and explain how our framework implement and optimize Eq. 2.

## 3 METHOD

### 3.1 OVERVIEW

Existing incremental learning methods primarily focus on enhancing models' adaptability to new classes, yet often neglect modeling the interpretability of decision-making processes. This leads to issues such as opaque decision-making, catastrophic forgetting of old knowledge, and insufficient understanding of new categories in complex and open environments. To address these challenges, we propose a Generative Explainable Class-Incremental Learning (GECL) framework. This

framework mitigates catastrophic forgetting by preserving knowledge of old classes through the generation of intrinsic structures within data distributions, while also providing interpretable, explicit reasoning chains for predictions. As illustrated in Fig. 2, our framework comprises three core modules. (a) A soft-label-guided visual enhancement module that highlights discriminative regions, reduces background interference, and improves visual input quality. (b) A generative semantic-visual extract module that employs large language models (LLMs) to generate structured attribute descriptions for new classes and extract hierarchical visual features as well. This module achieves class knowledge expansion while avoiding catastrophic forgetting caused by classification head expansion in discriminative models, requiring only lightweight parameter updates. (c) A distribution-aligned interpretable matching and inference module that formulates visual-semantic alignment as an entropy-regularized distribution matching problem. This generates region-attribute correspondences, enabling robust and interpretable predictions in incremental scenarios.

## 3.2 SOFT LABEL-GUIDED VISUAL ENHANCEMENT

Within the distribution matching framework, the quality of visual features directly impacts matching cost and classification performance. To enhance the discriminative power and alignability of regional features, we propose a soft-label-guided visual enhancement mechanism. This mechanism guides regional feature extraction using foreground probability distributions and gradient information, effectively suppressing background noise while improving semantic alignment capabilities for target regions. The core concept of this module involves structurally enhancing the input image to enable subsequent semantic-visual distribution matching within a feature space characterized by greater discriminative power, clearer boundaries, and reduce background interference.

We first employ gradient-based edge enhancement, calculating edge strength $E(x)$ via the Sobel operator to enhancing the structural features of prominent regions:

$$G_x = \begin{bmatrix} -1 & 0 & 1 \\ -2 & 0 & 2 \\ -1 & 0 & 1 \end{bmatrix} * I, G_y = \begin{bmatrix} -1 & -2 & -1 \\ 0 & 0 & 0 \\ 1 & 2 & 1 \end{bmatrix} * I, E = \sqrt{G_x^2 + G_y^2}. \quad (3)$$

Then, we apply adaptive image histogram equalization to local regions, dividing the image into multiple small areas and performing equalization independently on each to enhance texture contrast and suppress noise. Finally, we retain the original softmax output $\sigma(f_\theta(x))$ from the backbone network on the enhanced feature map as a foreground probability mask to guide the extraction of salient regions:

$$m(p) = \sigma(f_\theta(x)_p), \quad p \in \Omega. \quad (4)$$

The enhanced regional features can be represented as:

$$v_i(x) = \frac{1}{\sum_{p \in \mathcal{P}_i} m(p)} \sum_{p \in \mathcal{P}_i} m(p) \, \hat{v}_p \,, \quad (5)$$

where $\hat{v}_p$ denotes the pixel features after local enhancement processing.

## 3.3 GENERATIVE VISUAL-SEMANTIC MODELING FOR INTERPRETABLE CIL

Traditional incremental learning often relies on conditional probability for discriminative modeling, lacking separate modeling for each category. This approach fails to prevent task confusion and intra-task forgetting, making it difficult to extend to unseen categories. To address this, our approach is grounded in the framework of generative classification (Mackowiak et al., 2021). We model the joint probability $P(x, y)$ by marginalizing over a semantic concepts $A$:

$$P(x, y) = \sum_{A \in \mathcal{A}} P(y)P(A|y)P(x|A), \quad (6)$$

where $P(y)$ denotes to a uniform prior over classes, $P(A|y)$ denotes to a component to generate a set of semantic attributes $A$ for a class $y$. We implement this using a Large Language Model (LLM) as a powerful function approximator for this conditional distribution. The visual-semantic likelihood $P(x|A)$ represents the probability of the image $x$ given the semantics $A$:

$$P(x|A) \propto \exp(-\mathrm{DM}_\lambda(V(x), S(A))). \quad (7)$$

Here, a low distribution matching cost $\text{DM}_\lambda$ corresponds to a higher likelihood, signifying that the image $x$ is a good visual manifestation of the concepts in $A$. During inference, we approximate the MAP estimate by using the most representative semantic set $A_y$ for each class:

$$\hat{y} = \arg\max_y P(x, y) \approx \arg\min_y \text{DM}_\lambda(V(x), S(A_y)). \tag{8}$$

This formulation demonstrates that GECL is a generative model that performs classification by evaluating which class's generative process best explains the input image. This approach generates structured attribute sets for new classes without requiring any gradient updates, thereby avoiding parameter expansion in the classification head and effectively circumventing catastrophic forgetting.

Let $\mathcal{Y}$ denote the set of known classes. For any new class $y \in \mathcal{Y}$, we employ prompt templates to guide the LLMs in maximizing the posterior probability of generating semantic attributes under the constraint of prior knowledge:

$$A_y = \{a_j\}_{j=1}^n \sim P(A \mid \mathcal{P}_y; \Theta_{\text{LLM}}), \tag{9}$$

where $\mathcal{P}_y$ denotes the prompt template designed for category $y$, and $\Theta_{\text{LLM}}$ represents the language model parameters. The generated attribute tokens are mapped by the CLIP text encoder into a set of semantic attribute embeddings:

$$\mathcal{S}_y = \{f_{\text{text}}(a_i) \mid a_i \in A_y\}. \tag{10}$$

These embeddings will be injected into the knowledge base, jointly mapped with the semantic representations of all previous categories into the unified multimodal embedding space defined by CLIP. This approach eliminates the need for additional parameterized classifiers while preventing the forgetting of old knowledge due to parameter overwriting. For visual features, we employ a frozen CLIP visual encoder $f_{\text{image}}$ and combine it with a lightweight adapter module $g_\phi$ for domain adaptation. Given an input image $x$, its multi-scale visual feature representation is:

$$\mathcal{V} = \{f_{\text{image}}(x)^{(\text{global})}, f_{\text{image}}(x)_1^{(\text{local})}, \ldots, f_{\text{image}}(x)_m^{(\text{local})}\}, \tag{11}$$

where local features $f_{\text{image}}^{(\text{local})}$ are extracted by combining random sampling and center sampling. To achieve a transparent chain of reasoning, we also preserve the positional information of local visual features within the original image. This unified multimodal feature space not only supports incremental class expansion but also provides explicit semantic grounds for inference.

## 3.4 Interpretable Reasoning Module Based on Distribution Matching

After obtaining unified visual and semantic feature representations, the core challenge lies in establishing fine-grained correspondences between image regions and textual attributes. Simple similarity matching methods struggle to capture structured relationships and lack interpretable decision-making criteria. To address this, we model the task as an entropy-regularized optimal distribution alignment problem, transforming the association of visual regions with semantic attributes into the problem of finding the optimal match between two discrete distributions. We now define the entropy-regularized distribution matching operator $\text{DM}_\lambda$, show its closed-form parametrization, describe the iterative solver used in practice, and explain how the optimal coupling yields transparent reasoning.

Given an input image from which $m$ visual features $\mathcal{V} = \{\mathbf{v}_i\}_{i=1}^m$ are extracted, and a target class comprising $n$ semantic attribute features $\mathcal{S} = \{\mathbf{s}_j\}_{j=1}^n$, we define a cost matrix $\mathbf{C} \in \mathbb{R}^{m \times n}$ whose elements are defined by the cosine dissimilarity between feature pairs:

$$C_{ij} = 1 - \frac{\mathbf{v}_i \cdot \mathbf{s}_j}{\|\mathbf{v}_i\| \|\mathbf{s}_j\|}. \tag{12}$$

Let $p \in \Delta_m$ and $q \in \Delta_n$ be probability mass vectors over regions and attributes respectively. To obtain smooth and stable solutions, we introduce an entropy regularization term and formulate the following convex optimization problem:

$$\text{DM}_\lambda(V, S) = \min_{T \in \mathcal{U}(p,q)} \langle T, C \rangle - \lambda H(T), \tag{13}$$

where $\mathcal{U}(p, q) = \{T \in \mathbb{R}_+^{m \times n} : T\mathbf{1}_n = p, T^\top \mathbf{1}_m = q\}$ is the set of couplings with marginals $p$, $q$, $H(T) = -\sum_{i,j} T_{ij}(\log T_{ij} - 1)$ denotes the entropy of the coupling matrix $T$, and the regularization parameter $\lambda > 0$ is used to adjust the smoothness of the solution. Standard Lagrangian calculations (see Appendix B) show the minimizer of Eq. 13 has a diagonal scaling form:

$$T^\star = \text{diag}(u)\mathbf{K}\,\text{diag}(v), \quad u \in \mathbb{R}_{>0}^m,\ v \in \mathbb{R}_{>0}^n, \tag{14}$$

where $\mathbf{K}$ is the Gibbs kernel:

$$\mathbf{K} \in \mathbb{R}_{>0}^{m \times n}, \quad K_{ij} := \exp(-C_{ij}/\lambda), \tag{15}$$

and $\mathbf{u} \in \mathbb{R}^m$ and $\mathbf{v} \in \mathbb{R}^n$ are positive scaling vectors such that $T\mathbf{1}_n = p$ and $(T)^\top \mathbf{1}_m = q$. Enforcing the marginals yields the fixed-point iterations

$$\mathbf{u}^{(l+1)} = \mathbf{p} \oslash (\mathbf{K}\mathbf{v}^{(l)}), \quad \mathbf{v}^{(l+1)} = \mathbf{q} \oslash (\mathbf{K}^\top \mathbf{u}^{(l+1)}), \tag{16}$$

where $\oslash$ denotes element-wise division. This iterative process converges to a unique equilibrium point and exhibits numerical stability.(See Appendix B) Ultimately, the similarity between visual features and semantic features is measured by the regularization cost:

$$\hat{y}(x) = \arg \min_{y \in \mathcal{Y}^{(t)}} \mathrm{DM}_\lambda \left( V_\theta(x), S(y) \right). \tag{17}$$

By construction the optimal coupling $T^\star$ quantifies how much mass of region $i$ is assigned to attribute $j$. Higher-value elements $T_{ij}^\star$ in the matrix explicitly indicate which image regions jointly dominate the final classification decision with specific semantic attributes, therefore provide an explicit, quantitative region–attribute explanation: e.g., "Region 4 (0.83,0.75) — attribute 'thick white fur' (weight 0.018)". Extracting the top-k pairs $(i, j)$ with largest $T_{ij}^\star$ yields a compact, human-readable reasoning chain for each prediction.

To adapt visual features to the current task and align them with the semantic attribute distribution, we train a lightweight adapter using a cost-matrix-based classification loss. Given a training image x and its ground truth label y, the loss is defined as:

$$\mathcal{L}_{\mathrm{match}}(x, y) = \mathrm{DM}_\lambda \left( V_\theta(x), S(y) \right) + \epsilon \max_{y' \neq y} \left[ \gamma - \mathrm{DM}_\lambda(V_\theta(x), S(y')) \right]_+ \tag{18}$$

The first term represents the matching cost, while the second term is the contrastive loss, designed to widen the gap between positive classes and other categories. $\epsilon$ denotes the weight, and $\gamma$ is the hyperparameter. The overall training objective for adapter parameters $\theta$ at task $t$ is the empirical expectation of Eq. 18 plus the regularizer $\mathcal{R}(\theta)$ (see Eq. 2). Adapter parameters are updated by minimizing this loss, while the image encoder and text encoder remain frozen. During inference, the model only computes the cost matrix between the input image and the semantic sets of all classes in the knowledge base, eliminating the need for retraining. This approach prevents catastrophic forgetting and provides an explicit reasoning chain for each prediction.

The entropy-regularized optimal transport (DM) in GECL is not only a matching cost but also a mechanism for deriving reasoning. By enforcing a global distributional alignment between visual regions and class attributes, DM aggregates evidence across all regions rather than relying on isolated similarities. The entropy term encourages sparse yet non-binary couplings, allowing the model to weigh multiple plausible cues before converging on the most discriminative ones. The same coupling matrix that minimizes the DM cost also reveals which attributes support the final prediction, so prediction and explanation emerge from the same optimization rather than post-hoc interpretation.

## 4 EXPERIMENTS

### 4.1 DATASETS AND EVALUATION

**Datasets.** To comprehensively evaluate the performance of our method, we conduct experiments on traditional CIL scenarios and few-shot CIL scenarios using both natural image datasets and fine-grained datasets. For traditional CIL scenarios, we evaluate Tiny-ImageNet. Tiny-ImageNet is a subset of ImageNet comprising 200 categories, each containing 500 training images, 100 validation images, and 100 test images, all resized to 64×64. We train the first task in half 100 classes and split the other 100 classes into 5/10/20 tasks following Zhu et al. (2021a). For few-shot CIL scenarios,we evaluated two datasets: CUB200 (Catherine Wah & Belongie., 2011) and mini-ImageNet (Olga Russakovsky, 2015). CUB200 is a fine-grained bird classification dataset comprising 200 categories and 11,788 images. miniImageNet is a subset of ImageNet containing 50,000 training images across 100 selected classes. The base session includes 60 classes, while the remaining 40 classes are introduced over 8 incremental sessions with 5 new classes per session.

**Evaluation Metrics.** To evaluate performance across each dataset, we adopt classification accuracy (Acc) and performance drop (PD) as evaluation metrics. Acc assesses the model's discriminative capability and serves as the primary indicator for analyzing experimental results. PD measures the degree of accuracy decline between the final task and the initial task.

Table 1: Comparison of state-of-the-art methods on Tiny-ImageNet under the conventional CIL setting. "Avg" represents the averaged performance after training each task, and "Last" represents the performance on all test samples after training the last task.

| Type | Method | Accuracy on multiple tasks | | | | | |
|---|---|---|---|---|---|---|---|
| | | 5 tasks | | 10 tasks | | 20 tasks | |
| | | Avg | Last | Avg | Last | Avg | Last |
| Traditional models | EWC(Kirkpatrick et al., 2017b) | 19.01 | 6 | 15.82 | 3.79 | 12.35 | 4.73 |
| | LwF(Li & Hoiem, 2017) | 22.31 | 7.34 | 17.34 | 4.73 | 12.48 | 4.26 |
| | iCaRL(Rebuffi et al., 2017b) | 45.95 | 34.6 | 43.22 | 33.22 | 37.85 | 27.54 |
| | EEIL(Castro et al., 2018) | 47.17 | 35.12 | 45.03 | 34.64 | 40.41 | 29.72 |
| | UCIR(Hou et al., 2019) | 50.3 | 39.42 | 48.58 | 37.29 | 42.84 | 30.85 |
| | PASS(Zhu et al., 2021a) | 49.54 | 41.64 | 47.19 | 39.27 | 42.01 | 32.93 |
| | DyTox(Douillard et al., 2022b) | 55.58 | 47.23 | 52.26 | 42.79 | 46.18 | 36.21 |
| Discriminative models | Continual-CLIP(Thengane et al., 2022b) | 70.49 | 66.43 | 70.55 | 66.43 | 70.51 | 66.43 |
| | L2P(Wang et al., 2022e) | 83.53 | 78.32 | 76.37 | 65.78 | 68.04 | 52.40 |
| | DualPrompt(Wang et al., 2022c) | 85.15 | 81.01 | 81.38 | 73.73 | 73.45 | 60.16 |
| | InfLoRA(Liang & Li, 2024) | 83.20 | 77.85 | 82.75 | 77.65 | 81.60 | 77.40 |
| | CODA-Prompt(Smith et al., 2023b) | **85.91** | **81.36** | 82.8 | 75.28 | 77.43 | 66.32 |
| Generative models | GMM(Cao et al., 2024) | 83.42 | 76.98 | 82.49 | 76.51 | 81.70 | 76.03 |
| | GECL | 83.46 | 78.19 | **83.12** | **78.28** | **82.04** | **77.93** |

## 4.2 EXPERIMENTAL DETAILS

In this study, we employed MiniGPT-4 pre-trained projection layer checkpoint to generate attribute sets for each category. (Prompt: Which visual details are helpful for identifying a [CLS] in an image?) For both image and text encoding, we utilize a frozen CLIP (ViT-B/32) model. During image feature extraction, we set the central feature weight to 0.6. When generating semantic features, we cap the number of semantic features at 3–5 and set the entropy regularization parameter to 0.7. To optimize the learnable adapter, we employed Adam with a learning rate of 0.001. To mitigate overfitting, we introduce weight decay in the optimizer with a coefficient of 0.01. The momentum parameters are set to $\beta_1 = 0.9$, $\beta_2 = 0.98$. All experiments are conducted on two NVIDIA 3090 GPUs with 24GB VRAM each.

## 4.3 COMPARISON WITH STATE-OF-THE-ARTS

In both traditional CIL and few-shot CIL settings, we compare GECL against several state-of-the-art methods EWC(Kirkpatrick et al., 2017b), LwF(Li & Hoiem, 2017), iCaRL(Rebuffi et al., 2017b), EEIL(Castro et al., 2018), UCIR(Hou et al., 2019), PASS(Zhu et al., 2021a), DyTox(Douillard et al., 2022b),Continual-CLIP(Thengane et al., 2022b), L2P(Wang et al., 2022e), DualPrompt(Wang et al., 2022c), CODA-Prompt(Smith et al., 2023b), GMM(Cao et al., 2024), TOPIC(Tao et al., 2020c), Replay(Liu et al., 2022a), MetaFSCIL(Chi et al., 2022), NC-FSCIL(Yang et al., 2023), SACV(Song et al., 2023), F2M(Shi et al., 2021), CEC(Zhang et al., 2021b), FACL(Nema & Kurmi, 2025), Entropy-reg(Liu et al., 2022a), Icicle(Rymarczyk et al., 2023),InfLoRA (Liang & Li, 2024) across three challenging benchmarks, including discriminative approaches, pre-trained models, and prompt-based methods. Our approach achieved the best overall performance with an average accuracy of 80.28%, demonstrating a favorable balance between accuracy improvement, forgetting mitigation, and interpretability.

**Comparison on Tiny-imagenet.** Table 1 lists our performance results on the Tiny-ImageNet dataset. Specifically, GECL consistently outperforms all traditional methods such as LwF, iCaRL, and DyTox, highlighting the robustness of the generation process. Although GECL achieves slightly lower accuracy than DualPrompt and CODA-Prompt in short sequence settings (5 tasks), we attribute their advantages primarily to large-scale supervised pretraining on ImageNet-21K, which exhibits significant overlap with Tiny-ImageNet. However, our method demonstrates clear advantages in long sequence settings (10 and 20 tasks). This stems from the generative model's reduced reliance on the classification head, thereby mitigating task-specific bias and catastrophic forgetting. Furthermore, the comparison with GMM shows that our gains originate not merely from the generative methods, but from the effectiveness of generative semantic alignment.

**Comparison on CUB200.** We evaluate our method on the fine-grained CUB200 dataset, as shown in Table 2. We use Avg and Pd as key metrics for comparison with typical CIL methods. GECL

Table 2: Comparison of state-of-the-art methods on CUB200 under the few-shot CIL setting. Avg denotes mean accuracy across sessions, where higher values indicate better performance; PD represents the drop from the first to the last session, where lower values are preferred.

| Method | Accuracy in each task (%) ↑ | | | | | | | | | | | Avg ↑ | PD ↓ |
|---|---|---|---|---|---|---|---|---|---|---|---|---|---|
| | 0 | 1 | 2 | 3 | 4 | 5 | 6 | 7 | 8 | 9 | 10 | | |
| Icicle | 39.61 | 35.08 | 33.46 | 29.93 | 26.87 | 22.60 | 19.24 | 16.15 | 13.51 | 10.68 | 8.44 | 23.23 | 31.17 |
| iCaRL | 68.68 | 52.6 | 48.61 | 44.16 | 36.62 | 29.52 | 27.83 | 26.26 | 24.01 | 23.89 | 21.16 | 36.27 | 47.52 |
| EEIL | 68.68 | 53.63 | 47.91 | 44.20 | 36.30 | 27.46 | 25.93 | 24.70 | 23.95 | 24.13 | 22.11 | 36.27 | 46.57 |
| TOPIC | 68.68 | 62.49 | 54.81 | 49.99 | 45.25 | 41.40 | 38.35 | 35.36 | 32.22 | 28.31 | 26.28 | 43.92 | 42.42 |
| Replay | 75.90 | 72.14 | 68.64 | 63.76 | 62.58 | 59.11 | 57.82 | 55.89 | 54.92 | 53.58 | 52.39 | 61.52 | 23.51 |
| MetaFSCIL | 75.90 | 72.41 | 68.78 | 64.78 | 62.96 | 59.99 | 58.30 | 56.85 | 54.78 | 53.82 | 52.64 | 61.93 | 23.26 |
| NC-FSCIL | 80.45 | 75.98 | 72.30 | 70.28 | 68.17 | 65.16 | 64.43 | 63.25 | 60.66 | 60.01 | 59.44 | 67.28 | 21.01 |
| SAVC | 81.55 | 77.92 | 74.95 | 70.21 | 69.96 | 67.02 | 66.16 | 65.30 | 63.84 | **63.15** | **62.50** | 69.35 | **19.35** |
| F2M | 81.07 | 78.16 | 75.57 | 72.89 | 70.86 | 68.17 | 67.01 | 65.26 | 63.36 | 61.76 | 60.26 | 69.49 | 20.81 |
| InfLoRA | **83.71** | **81.97** | 78.94 | 76.47 | 75.25 | 74.10 | 70.28 | 67.73 | 63.23 | 61.10 | 59.17 | 72.00 | 24.54 |
| GECL(Ours) | 83.52 | 81.33 | **79.33** | **76.83** | **75.67** | **74.67** | **70.89** | **68.57** | **64.58** | 62.00 | 59.97 | **72.49** | 23.55 |

Table 3: Comparison of state-of-the-art methods on mini-ImageNet under the few-shot CIL setting. Avg denotes mean accuracy across sessions, where higher values indicate better performance; PD represents the drop from the first to the last session, where lower values are preferred. ∗ indicates our re-implementation based on PILOT(Sun et al., 2025)

| Method | Accuracy in each task (%) ↑ | | | | | | | | | Avg ↑ | PD ↓ |
|---|---|---|---|---|---|---|---|---|---|---|---|
| | 0 | 1 | 2 | 3 | 4 | 5 | 6 | 7 | 8 | | |
| iCaRL | 61.31 | 46.32 | 42.94 | 37.63 | 30.49 | 24 | 20.89 | 18.8 | 17.21 | 33.29 | 44.1 |
| EEIL | 61.31 | 46.58 | 44.00 | 37.29 | 33.14 | 27.12 | 24.10 | 21.57 | 19.58 | 34.97 | 41.73 |
| UCIR | 61.31 | 47.8 | 39.31 | 31.91 | 25.68 | 24.35 | 18.67 | 17.24 | 14.17 | 31.16 | 47.14 |
| TOPIC | 61.31 | 50.09 | 45.17 | 41.16 | 37.48 | 35.52 | 32.19 | 29.46 | 24.42 | 39.64 | 36.89 |
| Replay | 71.84 | 67.12 | 63.21 | 59.77 | 57.01 | 53.95 | 51.55 | 49.52 | 48.21 | 58.02 | 23.63 |
| CEC | 72.00 | 66.83 | 62.97 | 59.43 | 56.70 | 53.73 | 51.19 | 49.24 | 47.63 | 57.75 | 24.37 |
| F2M | 72.05 | 67.47 | 63.16 | 59.70 | 56.71 | 53.77 | 51.11 | 49.21 | 47.84 | 57.89 | 24.21 |
| MetaFSCIL | 72.04 | 67.94 | 63.77 | 60.29 | 57.58 | 55.16 | 52.90 | 50.79 | 49.19 | 58.85 | 22.85 |
| SAVC | 81.12 | 76.14 | 72.43 | 68.92 | 66.48 | 62.95 | 59.92 | 58.39 | 57.11 | 67.05 | 24.01 |
| FACL | 86.68 | 81.49 | 76.65 | 72.65 | 69.71 | 66.02 | 63.08 | 61.17 | 59.48 | 70.77 | 27.20 |
| Entropy-reg | 71.84 | 67.12 | 63.21 | 59.77 | 57.01 | 53.95 | 51.55 | 49.52 | 48.21 | 58.02 | 23.63 |
| L2P ∗ | 93.57 | 87.86 | 81.84 | 75.16 | 70.51 | 65.67 | 62.11 | 59.61 | 56.28 | 72.51 | 37.29 |
| DualPrompt ∗ | 93.12 | 86.31 | 81.65 | 77.24 | 71.43 | 66.57 | 52.32 | 59.16 | 56.94 | 71.64 | 36.18 |
| CODA-Prompt ∗ | **95.42** | 88.89 | 82.76 | 77.92 | 74.77 | 70.54 | 66.18 | 63.47 | 61.22 | 75.69 | 34.20 |
| InfLoRA | 94.87 | 91.53 | 85.61 | 83.24 | 81.95 | 80.88 | 79.43 | 77.86 | 74.12 | 82.39 | 20.75 |
| GMM | 89.35 | 88.40 | 86.11 | **85.07** | **83.61** | 81.35 | 78.97 | 77.34 | 75.18 | 82.82 | **14.17** |
| GECL(Ours) | 95.19 | **93.33** | **86.67** | 84.44 | 83.27 | **82.67** | **81.35** | **80.19** | **77.00** | **84.90** | 16.33 |

achieves an average accuracy of 72.49%, representing a 36.22% improvement over traditional CIL methods (e.g., iCaRL's 36.27%). In contrast, the interpretable method Icicle achieves only 23.23% average accuracy, significantly underperforming our approach. Although SAVC achieves higher accuracy on the last two tasks and InfLoRA behaves better on the first two tasks, our method consistently outperforms on subsequent tasks due to its explicit distribution matching between visual regions and semantic attributes, enabling sustained scalability and interpretability. These results validate GECL's generalization capability for learning incremental classes under limited data scenarios.

**Comparison on mini-ImageNet.** Table 3 presents our comparison results against several representative baselines on the mini-ImageNet few-shot dataset. We report results using the average accuracy

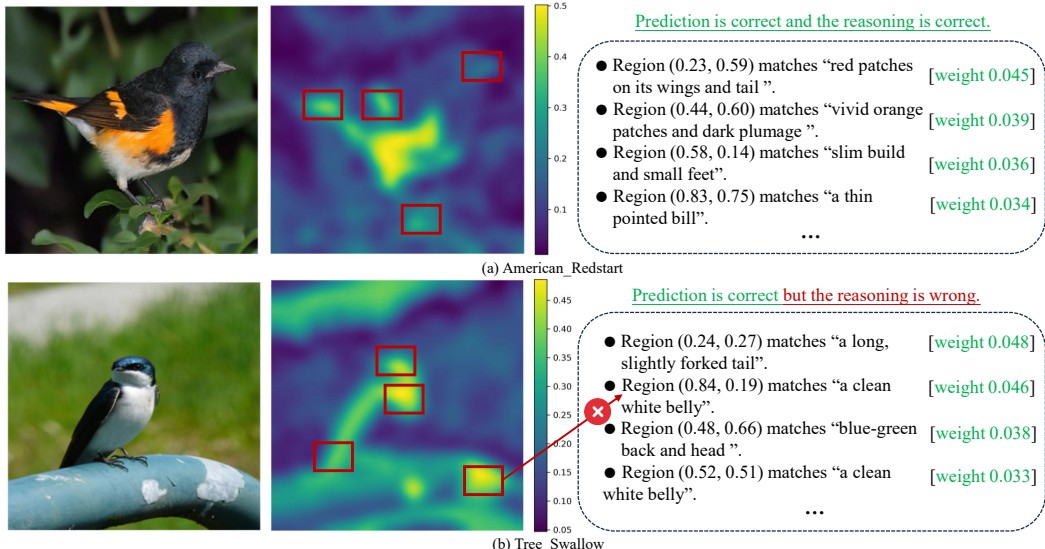

Figure 3: Visualization of transparent reasoning chain.The models in Figures (a) and (b) both made correct predictions, but the reasoning in Figure b was incorrect. This demonstrates that our approach can provide transparent chains of reasoning, thereby enhancing the model's credibility.

across all tasks (Avg) and the final performance drop (PD). Our proposed method GECL achieves the best overall performance, with an average accuracy of 84.90% and a PD of only 16.33%. This demonstrates both strong adaptability to new tasks and exceptional knowledge retention capabilities. Compared to classical discriminative methods like iCaRL and UCIR, our approach achieves over 40% performance improvement. Furthermore, GECL achieves an 8% accuracy gain over the state-of-the-art prompt-based baseline CODA-Prompt while maintaining a smaller PD. Although our initial performance on the first task is slightly below CODA-Prompt, GECL consistently outperforms it in subsequent stages by effectively mitigating catastrophic forgetting while enhancing salient visual features. In contrast, GECL strikes a balance between stability and interpretability, making it highly effective for few-shot class-incremental learning.

### 4.4 ABLATION STUDIES

To better understand the contribution of each component in our GECL framework, we conduct an ablation study on three benchmark datasets, as reported in Table 4. Starting from the baseline model, incorporating the optimal matching module (DM) leads to a notable performance improvement, e.g., from 54.34% to 75.96% on Tiny-ImageNet, indicating that fine-grained alignment between visual and semantic features effectively mitigates rep-

Table 4: Performance comparison on different datasets.

| Module | Datasets | | |
|---|---|---|---|
| | Tiny-ImageNet | CUB200 | mini-ImageNet |
| Baseline | 54.34 | 49.38 | 70.62 |
| Baseline+DM | 75.96 | 63.06 | 77.51 |
| Baseline+SL | 68.57 | 57.28 | 74.95 |
| Baseline+DM+SL | 83.04 | 71.41 | 82.53 |
| **GECL(Ours)** | **83.46** | **72.49** | **84.90** |

resentation mismatch. The addition of soft-label guided enhancement (SL) also brings consistent gains across datasets, demonstrating that soft label helps preserve semantic consistency during incremental updates. Combining both DM and SL achieves further improvements, confirming their complementary effects. Finally, our full model, which integrates adaptive feature weighting on top of DM and SL, achieves the best results on all datasets (83.46%, 72.49%, and 84.90%), highlighting the effectiveness of dynamically balancing global and local features for robust and interpretable incremental learning. More ablation Analysis see Appendix D and Fig. 4.

### 4.5 VISUALIZATIONS

Fig. 3 presents visualization results on the CUB200 dataset. Both models in (a) and (b) produce correct predictions; however, their reasoning chains reveal a key difference. The prediction in (a)

is based on correct evidence, whereas in (b) the model incorrectly matches a white patch on the background railing with the "clean white belly" attribution of the Tree Swallow. This demonstrates that a model can achieve correct predictions for the wrong reasons. Our method addresses this issue by not only producing accurate predictions but also providing transparent and interpretable reasoning, thereby enhancing the trustworthiness of model outputs in critical scenarios.

### 4.6 QUANTITATIVE EVALUATION OF EXPLAINABILITY

While GECL provides explicit region–attribute reasoning chains through the coupling matrix $T$, we further conduct a quantitative analysis to evaluate the spatial faithfulness of these explanations. Following Icicle(Rymarczyk et al., 2023), we adopt the IoU metric derived from the Top-K semantic evidence inside the optimal transport reasoning path. Because GECL generates free-form natural-language attributes, we establish

Table 5: IoU on CUB200 dataset.

| Method | Task 1 | Task 2 | Task 3 | Mean |
|--------|--------|--------|--------|------|
| Finetuning | 0.115 | 0.149 | 0.260 | 0.151 |
| EWC | 0.192 | 0.481 | 0.467 | 0.334 |
| LWF | 0.221 | 0.193 | 0.077 | 0.188 |
| LWM | 0.332 | 0.312 | 0.322 | 0.325 |
| Icicle | 0.705 | 0.753 | 0.742 | 0.728 |
| **GECL(Ours)** | **0.752** | **0.781** | **0.769** | **0.767** |

a deterministic mapping from each attribute phrase to the 15 CUB part categories. The mapping is constructed by combining phrase-level lexical rules with CLIP similarity to the 312 official CUB attributes. This allows us to translate each DM-selected attribute–patch pair into a predicted anatomical part, enabling part-level IoU computation. We also provide a human evaluation of the explainability of GECL inspired by FunnyBirds (Hesse et al., 2023). Details see Appendix C.

Table. 5 reports the IoU scores on CUB200 dataset. Traditional continual learning methods including Finetuning, EWC, LWF, and LWM obtain low IoU scores, indicating that their predictions often rely on visually misaligned or semantically inconsistent regions, despite sometimes achieving competitive classification accuracy. Icicle, an interpretable CIL method based on prototype–part alignment, significantly improves the IoU to 0.728 on average, reflecting its use of part-based representations.

GECL achieves the highest explainability across all tasks, with a mean IoU of 0.767, outperforming ICICLE by +3.9%. This improvement stems from GECL's DM-based visual–semantic alignment, which grounds the classification decision on meaningful attributes and their corresponding spatial regions. The consistent improvement across tasks demonstrates that GECL's explanations remain stable even under long-sequence incremental learning, validating that the structured reasoning chain generated by GECL is spatially faithful, semantically coherent, and robust to task progression.

## 5 CONCLUSION

**Summary.** In this work, we introduce Generative Explainable Class-Incremental Learning (GECL), a novel framework that integrates generative semantic modeling with interpretable reasoning for incremental learning. Unlike traditional discriminative methods that suffer from catastrophic forgetting and opaque decision processes, GECL preserves prior knowledge by generating structured semantic attributes and aligning them with visual regions through entropy-regularized optimal transport. This enables the model to expand knowledge continuously while producing transparent attribute–region reasoning chains for each prediction. Extensive experiments on both natural and fine-grained datasets demonstrate that GECL achieves a favorable balance between accuracy, forgetting resistance, and interpretability. We believe this work provides a promising step toward safe, transparent, and scalable continual learning.

**Limitations and Future Work.** Our approach relies on the selection of LLMs and the quality of generated semantic attributes, but LLMs occasionally output background or entirely irrelevant information. GECL is not directly applicable to non-visual continual learning settings (e.g., text-only LLM continual learning) without redefining the grounding mechanism, which we identify as a promising direction for future work. Future work could also focus on integrating different LLMs and developing improved prompting and information filtering mechanisms to enhance the relevance of semantic attributes.

## A REPRODUCIBILITY STATEMENT

We are committed to ensuring the reproducibility of our work. To this end, we provide the following:

**Source Code:** We provide the core code for our method, GECL, in the following repository: `https://anonymous.4open.science/r/ICLR-3FCD/`

**Datasets and Evaluation Metrics:** All datasets used in our experiments are publicly available and described in the paper. We clearly outline the evaluation metrics used and explain the motivations for choosing them in Section 4.1.

**Theoretical Contributions:** All assumptions presented in Appendix B are clearly and formally stated. Proofs for all novel claims are included. Each lemma and theorem follows a standard mathematical derivation procedure, and intermediate results are explicitly stated to avoid ambiguity.

**Experimental Setup:** We report the hyperparameters, training schedule, and optimization settings in Section4.2. We specify the random seeds used in the experiments to ensure consistent reproducibility. By reporting these details, we aim to enable independent researchers to replicate and improve upon the results reported in our paper.

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

## A  THE USE OF LARGE LANGUAGE MODELS

Large language models (LLMs) are employed in research to generate text descriptions, leveraging semantic information to achieve interpretable visual-language distribution alignment. The role of LLMs in research conception, writing, data analysis, result interpretation, or the conduct of the study itself is strictly limited to linguistic refinement, grammatical correction, and enhancing the clarity and fluency of the paper. All intellectual contributions, including problem formulation, methodology development, and conclusion derivation, are solely the work of human authors.

## B  LEMMAS AND PROOFS

### B.1  LEMMAS

**Lemma 1.** *Assume $C \in \mathbb{R}^{m \times n}$ is finite and $\lambda > 0$. Let $p \in \Delta_m, q \in \Delta_n$ have strictly positive entries. Then the optimization problem Eq. 13 admits a unique minimizer $T^\star \in \mathcal{U}(p, q)$. Moreover $T^\star$ has the diagonal-scaling representation Eq. 14 with strictly positive $u, v$.*

**Lemma 2.** *Let $K = \exp(-C/\lambda)$ and assume $K_{ij} > 0$ for all $i, j$. The iterative matrix balancing updates Eq. 16, initialized with any positive vector $v^{(0)} > 0$, generate positive $u^{(k)}, v^{(k)}$ and couplings $T^{(k)} = \mathrm{diag}(u^{(k)}) K \mathrm{diag}(v^{(k)})$ that converge to the unique minimizer $T^\star$ of Eq. 13. The sequence of objective values is monotone decreasing and bounded below.*

### B.2  PROOFS

*Proof.* Consider the objective

$$J(T) := \langle T, C \rangle - \lambda H(T) = \sum_{i,j} T_{ij} C_{ij} + \lambda \sum_{i,j} T_{ij} \log T_{ij} - \lambda \sum_{i,j} T_{ij}. \tag{19}$$

The feasible set $\mathcal{U}(p, q)$ is a non-empty, compact convex subset of $\mathbb{R}_+^{m \times n}$ (since $p, q > 0$ and $m, n$ finite). Inside the positive otrhant the function $T \mapsto \sum T_{ij} \log T_{ij}$ is strictly convex. Thus $J(T)$ is strictly convex on the interior of $\mathcal{U}(p, q)$, so any minimizer is unique (if it exists in the interior). Because $J$ is continuous on the compact feasible set, a minimizer exists. Combining existence and strict convexity gives uniqueness.

To obtain the diagonal scaling parametrization, gorm the Lagrangian with multipliers $\alpha \in \mathbb{R}^m$ and $\beta \in \mathbb{R}^n$ enforcing the marginal constraints:

$$\mathcal{L}(T, \alpha, \beta) = \sum_{i,j} T_{ij} C_{ij} + \lambda \sum_{i,j} T_{ij} \log T_{ij} - \lambda \sum_{i,j} T_{ij} + \sum_i \alpha_i (p_i - \sum_j T_{ij}) + \sum_j \beta_j (q_j - \sum_i T_{ij}). \tag{20}$$

Setting $\partial \mathcal{L}/\partial T_{ij} = 0$ for each $i, j$ yields

$$C_{ij} + \lambda(\log T_{ij} + 1) - \lambda - \alpha_i - \beta_j = 0 \implies \log T_{ij} = -\frac{C_{ij}}{\lambda} + a_i + b_j, \tag{21}$$

where $a_i = -\frac{\alpha_i}{\lambda}$a and $b_j = -\frac{\beta_j}{\lambda}$. Exponentiating gives

$$T_{ij} = e^{a_i} e^{-C_{ij}/\lambda} e^{b_j} = u_i K_{ij} v_j, \tag{22}$$

which is the diagonal scaling form Eq. 14 with $u_i = e^{a_i}, v_i = e^{b_j} > 0$. The scaling vectors $u, v$ are determined uniquely by solving the marginal constraints $T\mathbf{1} = p, T^\top \mathbf{1} = q$. Because the solution exists and is unique in the positive orthant (compactness and strict convexity), Lemma 1 follows. $\square$

*Proof.* We prove that the iterative updates in Eq. 16 converge to the unique minimizer $T^\star$.

**KL-projection equivalence.**  Writing $K$ as in Eq. 14, we observe that the regularized transport objective can be rewritten as:

$$\langle T, C \rangle - \lambda H(T) = \lambda \sum_{i,j} T_{ij} \log \frac{T_{ij}}{K_{ij}} - \lambda \sum_{i,j} K_{ij} + \text{const.} \tag{23}$$

Thus, minimizing $J(T)$ over the transportation polytope $\mathcal{U}(p,q)$ is equivalent to computing the I-projection (KL projection) of the positive matrix $K$ onto $\mathcal{U}(p,q)$:

$$T^\star = \arg \min_{T \in \mathcal{U}(p,q)} \mathrm{KL}(T \parallel K). \tag{24}$$

This formulation is standard and implies that the iterative proportional fitting procedure (IPFP), also known as iterative scaling, i.e., alternately normalizing rows and columns, is a natural solver: each scaling step enforces one marginal constraint exactly while monotonically decreasing the KL divergence to $K$.

**Monotone decrease and boundedness.** Let $T^{(k)}$ denote the coupling after $k$ half-steps (either row or column normalization). It can be shown, by convexity of the KL divergence along coordinate projections, that:

$$\mathrm{KL}\big(T^{(k+1)} \parallel K\big) \ \leq \ \mathrm{KL}\big(T^{(k)} \parallel K\big), \tag{25}$$

and hence the sequence $\{\mathrm{KL}(T^{(k)} \parallel K)\}$ is non-increasing and bounded below by $0$. Therefore, the sequence converges to some limit value $L^\star \geq 0$.

**Convergence to the projection.** The compactness of $\mathcal{U}(p,q)$ ensures that any sequence $\{T^{(k)}\}$ has accumulation points. By standard results on alternating I-projections onto convex sets (see, e.g., Csiszar (1975)), any accumulation point of $\{T^{(k)}\}$ attains the minimal KL value $L^\star$ and therefore must coincide with the unique KL projection $T^\star$. Hence, the entire sequence converges to $T^\star$.

$\square$

### B.3 Reformulating CIL as Semantic-Visual Distribution Matching

In traditional class-incremental learning, models typically perform discriminative classification through conditional probability $P(y|x)$. However, this approach faces catastrophic forgetting in incremental learning scenarios. We propose to reformulate CIL as a semantic-visual distribution matching problem, with the following theoretical foundation:

**Definition 1.** *(Semantic-Visual Distribution Matching). Given visual feature distribution $\mathcal{V}(x)$ and semantic feature distribution $\mathcal{S}(y)$, we define the matching score for class $y$ as:*

$$\mathrm{Match}(x,y) = -\mathrm{DM}_\lambda(\mathcal{V}(x), \mathcal{S}(y)) \tag{26}$$

*where $DM_\lambda$ is the entropy-regularized optimal transport distance.*

**Lemma 3.** *The dual form of the entropy-regularized optimal transport problem is:*

$$\mathrm{OT}_\lambda(\mathcal{V}, \mathcal{S}) = \max_{u,v} \langle u,p \rangle + \langle v,q \rangle - \lambda \langle e^{u/\lambda}, K e^{v/\lambda} \rangle \tag{27}$$

*where $K = \exp(-C/\lambda)$ is the Gibbs kernel matrix.*

*Proof.* Consider the primal problem:

$$\min_{T \in \mathcal{U}(p,q)} \langle T, C \rangle - \lambda H(T) \tag{28}$$

Introduce Lagrange multipliers $u, v$:

$$\mathcal{L}(T,u,v) = \langle T, C \rangle - \lambda H(T) + \langle u, p - T\mathbf{1} \rangle + \langle v, q - T^\top \mathbf{1} \rangle \tag{29}$$

Differentiate with respect to $T$ and set to zero:

$$\frac{\partial \mathcal{L}}{\partial T_{ij}} = C_{ij} + \lambda(\log T_{ij} + 1) - u_i - v_j = 0 \tag{30}$$

Solving yields:

$$T_{ij} = \exp\left(\frac{u_i + v_j - C_{ij}}{\lambda} - 1\right) = \frac{1}{e} \exp\left(\frac{u_i}{\lambda}\right) \exp\left(-\frac{C_{ij}}{\lambda}\right) \exp\left(\frac{v_j}{\lambda}\right) \tag{31}$$

Substituting into the marginal constraints gives the dual problem. $\square$

**Theorem 1.** *Let $\mathcal{H}$ be the hypothesis space of our model, and $\mathcal{R}(h) = \mathbb{E}_{(x,y)\sim\mathcal{D}}[\ell(h(x), y)]$ be the expected risk. For any hypothesis $h$ that makes predictions via distribution matching:*

$$h(x) = \arg\min_{y\in\mathcal{Y}} \mathrm{DM}_\lambda(\mathcal{V}(x), \mathcal{S}(y)) \tag{32}$$

*the classification risk is bounded by:*

$$\mathcal{R}(h) \leq \mathbb{E}_{(x,y)\sim\mathcal{D}}[\mathrm{DM}_\lambda(\mathcal{V}(x), \mathcal{S}(y))] + \mathcal{C}_1\epsilon_{\mathrm{sem}} + \mathcal{C}_2\lambda + \mathcal{C}_3\sqrt{\frac{\log(1/\delta)}{2n}} \tag{33}$$

*where $\epsilon_{sem}$ is the semantic approximation error, $\lambda$ is the entropy regularization parameter.*

*Proof.* We prove this in steps:

Let $\mu_y$ be the true visual feature distribution for class $y$, and $\nu_y$ be the true semantic distribution. By the triangle inequality for Wasserstein distance:

$$W_1(\mathcal{V}(x), \mathcal{S}(y)) \leq W_1(\mathcal{V}(x), \mu_y) + W_1(\mu_y, \nu_y) + W_1(\nu_y, \mathcal{S}(y))$$
$$\leq W_1(\mathcal{V}(x), \mu_y) + W_1(\mu_y, \nu_y) + \epsilon_{\mathrm{sem}} \tag{34}$$

Following Cuturi (2013), we have:

$$\mathrm{DM}_\lambda(\mathcal{V}, \mathcal{S}) \leq W_1(\mathcal{V}, \mathcal{S}) + \lambda\log(NM) \tag{35}$$

where $N, M$ are the support sizes of $\mathcal{V}$ and $\mathcal{S}$ respectively. Classification error occurs when there exists $y' \neq y$ such that:

$$\mathrm{DM}_\lambda(\mathcal{V}(x), \mathcal{S}(y)) > \mathrm{DM}_\lambda(\mathcal{V}(x), \mathcal{S}(y')) \tag{36}$$

Define the margin function:

$$\gamma(x, y) = \min_{y'\neq y}[\mathrm{DM}_\lambda(\mathcal{V}(x), \mathcal{S}(y')) - \mathrm{DM}_\lambda(\mathcal{V}(x), \mathcal{S}(y))] \tag{37}$$

Then the 0-1 loss satisfies:

$$\ell(h(x), y) \leq 1_{\gamma(x,y)\leq 0} \tag{38}$$

Using Rademacher complexity analysis, with hypothesis class complexity $\mathfrak{R}_n(\mathcal{H})$, for any $\delta > 0$, with probability at least $1 - \delta$:

$$\mathcal{R}(h) \leq \frac{1}{n}\sum_{i=1}^{n}\phi(\gamma(x_i, y_i)) + \frac{2L\mathfrak{R}_n(\mathcal{H})}{\gamma} + \sqrt{\frac{\log(1/\delta)}{2n}} \tag{39}$$

where $\phi$ is a surrogate loss function and $L$ is the Lipschitz constant. We finally obtain:

$$\mathcal{R}(h) \leq \mathbb{E}[\mathrm{DM}_\lambda(\mathcal{V}(x), \mathcal{S}(y))] + \mathcal{C}_1\epsilon_{\mathrm{sem}} + \mathcal{C}_2\lambda + \mathcal{C}_3\sqrt{\frac{\log(1/\delta)}{2n}} \tag{40}$$

where constants $\mathcal{C}_1, \mathcal{C}_2, \mathcal{C}_3$ depend on the Lipschitz properties of the visual and semantic encoders. □

## C EXPLAINABILITY EVALUATION

### C.1 INTRODUCTION OF THE IOU METRIC

GECL performs classification through an entropy-regularized Optimal Matching (DM) between visual regions and semantic attributes. The optimal coupling matrix $T^\star$ explicitly encodes the evidence used in the prediction. We design an explainability metric which measures how well this evidence aligns spatially with ground-truth CUB parts.

For a test image $x$, we extract the Top-$K$ largest entries of $T^\star$:

$$\mathcal{P} = \{(i,j) \mid T_{ij}^\star \text{ in Top-}K\},$$

where $(i, j)$ denotes visual patch $i$ supporting attribute $a_j$. These patches constitute the causal evidence in the model's reasoning chain. Each visual patch corresponds to a spatial region $R_i(x)$, therefore the model-evidence mask is:

$$M(x) = \bigcup_{(i,j)\in\mathcal{P}} R_i(x).$$

Each semantic attribute $a_j$ is mapped to one or more of the 15 CUB parts:

$$\mathcal{A}(a_j) \subseteq \{1, \ldots, 15\},$$

using rule-based matching or semantic similarity. The predicted spatial mask for part $p$ aggregates all evidence patches linked to attributes associated with $p$:

$$\hat{G}_p(x) = \bigcup_{(i,j)\in\mathcal{P}:p\in\mathcal{A}(a_j)} R_i(x).$$

Given ground-truth part mask $G_p(x)$, we compute:

$$\text{IoU}_p(x) = \frac{|\hat{G}_p(x) \cap G_p(x)|}{|\hat{G}_p(x) \cup G_p(x)|}.$$

The final IoU is averaged across all parts and all images.

This metric evaluates the spatial faithfulness of GECL's DM-based reasoning chain, directly measuring whether the visual evidence supporting semantic attributes aligns with true object parts.

## C.2 CONSTRUCTION OF THE ATTRIBUTE–PART MAPPING TABLE

GECL produces over 1,500 natural-language attributes, many of which contain complex semantics (e.g., "a black head and neck, with a white face and black bill"). To align these free-form descriptions with the standardized CUB part ontology, we create a comprehensive attribute–part mapping table. Each attribute phrase is normalized (lowercased, punctuation removed) and matched to the closest predefined phrase pattern or keyword group associated with one or more CUB parts. If no direct match is found, we ues CLIP text–text similarity: each GECL phrase is matched against the 312 official CUB attributes, and we inherit the part label from the nearest attribute.The resulting mapping associates each attribute phrase with one or more of the 15 CUB parts (e.g., crown, throat, left wing, belly). This table is used only for evaluation, and it does not influence training.

## C.3 HUMAN EVALUATION OF EXPLAINABILITY

To complement the IoU metric, we conducted a controlled human study to assess the perceived explainability of GECL reasoning chains. Five graduate students with basic computer vision background (non-experts in ornithology) participated in the evaluation. Each annotator was independently shown a set of 150 randomly sampled test images, each accompanied by GECL's predicted label and its Top-K region–attribute reasoning chain.

Table 6: Human evaluation on CUB200.

| Metric | Score |
|---|---|
| Faithfulness | 4.28 ± 0.16 |
| Clarity | 4.21 ± 0.14 |
| Part Consistency | 4.17 ± 0.15 |
| Trust / Usefulness | 4.31 ± 0.12 |

For each image, inspired by FunnyBirds(Hesse et al., 2023), annotators rated GECL's explanation according to four criteria, using a 5-point Likert scale (1 = strongly disagree, 5 = strongly agree). **Faithfulness** denotes to whether the highlighted patch genuinely contains the visual evidence described by the corresponding attribute. **Clarity** denotes to whether the region–attribute pairs are easy to understand and form a coherent explanation. **Part Consistency** denotes to whether the semantic attribute reasonably matches the part of the bird indicated by the highlighted region (according to the CUB part taxonomy). **Trust and Usefulness** denote to whether the explanation increases the annotator's confidence in the correctness of the model's prediction. Each annotator completed three independent runs (different random samples per run). Final scores in Table 8 are averaged across annotators and runs.

Human evaluation confirms that GECL's region–attribute reasoning chains are not only machine-interpretable but also perceptually meaningful and trustworthy to human users. The high faithfulness and part-consistency scores suggest that the DM-derived alignment between visual patches and semantic attributes generally corresponds to correct anatomical regions on the bird, validating the effectiveness of GECL's semantic grounding. Moreover, the consistently high trust scores indicate that GECL's explanations help users understand why a prediction is made, reducing the "black-box" gap typical in continual learning systems. These results demonstrate that GECL achieves human-validated explainability, complementing its quantitative IoU improvements and reinforcing its suitability for real-world incremental learning scenarios where transparency is essential.

# D ABLATION STUDIES

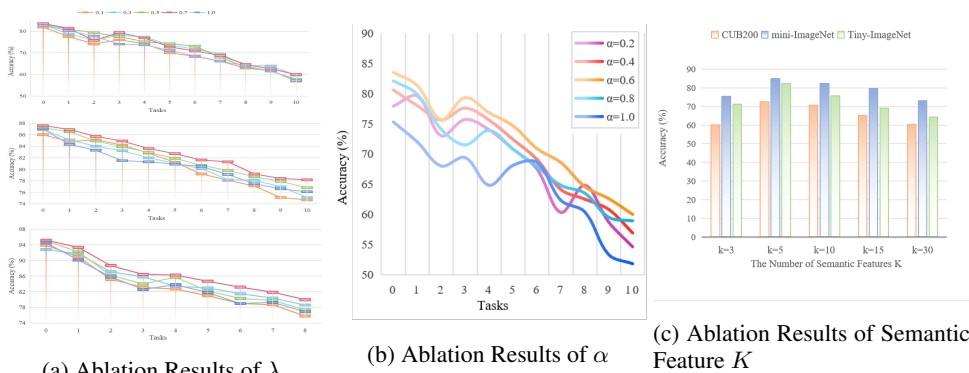

(a) Ablation Results of $\lambda$.

(b) Ablation Results of $\alpha$

(c) Ablation Results of Semantic Feature $K$

Figure 4: Results of Ablation Studies

**Ablation of Entropy Regularization $\lambda$ .** We first analyze the impact of the entropy regularization parameter $\lambda$ on model performance. Results indicate that when $\lambda$ is too small (e.g., 0.1), the optimal transport matrix tends toward sparsity, leading to overly aggressive cross-modal matching that causes training instability and significant forgetting in later stages. Conversely, when $\lambda$ is too large (e.g., 1.0), the matching distribution becomes excessively smooth, weakening the discriminative capability between regions and semantic attributes. In contrast, a moderate $\lambda = 0.7$ consistently achieves optimal results across CUB200, Mini-ImageNet, and Tiny-ImageNet, demonstrating that reasonable regularization strikes a balance between "stability" and "discriminative power." This trend progressively amplified during the incremental phase, fully validating our method's robustness to $\lambda$ settings.

**Ablation of Semantic Features Length.** We conducte comparisons under varying numbers of semantic features. Results indicate that insufficient attributes (e.g., $k = 3$) lead to inadequate semantic expressiveness, impairing the classification of new classes. Conversely, excessive attributes (e.g., $k = 30$) offer broader coverage but introduce semantic redundancy and noise, resulting in degraded performance during the accumulation phase. The optimal configuration $k = 5$ achieves the highest average accuracy and lowest forgetting rate across all three datasets, demonstrating that moderate and high-quality semantic descriptions are crucial for incremental learning. This finding further underscores the role of generative semantic augmentation in combating forgetting—supporting model scalability and stability through refined, non-redundant semantic modeling.

**Ablation of Sampling Weights.** We further examine the weight $\alpha$ between local random sampling and center sampling. Experiments reveal that when $\alpha$ is too low (random-biased), the feature representations contain excessive noise, weakening the discriminative power of attribute alignment. Conversely, when $\alpha$ is too high (center-biased), the model over-relies on the target subject while neglecting crucial local fine-grained information. Both extremes lead to performance degradation during long-term incremental processes. In contrast, $\alpha = 0.6$ (central-biased yet retaining partial randomness) consistently outperformed other configurations, validating that a hybrid sampling strategy of "global consistency + local diversity" is more suitable for incremental alignment and forgetting resistance. This indicates that the sampling mechanism itself is a crucial factor influencing model stability.

**Noise-injection Experiments for LLMs** To quantitatively verify robustness, as shown in Table. 7, we conducted noise-injection experiments where 10–50% of the attributes were randomly replaced. GECL retained 69.6% (–2.0%) accuracy and 0.738 (–0.029) IoU under 20% noise, confirming that classification and interpretability remain robust under realistic imperfect semantic conditions.

Table 7: Attribute noise injection (random attribute replacement). Results on CUB (10-task setting).

| Noise | Accuracy (%) | ΔAcc (%) | IoU | ΔIoU |
|---|---|---|---|---|
| 0% | 71.6 | 0.0 | 0.767 | 0.000 |
| 10% | 70.8 | -0.8 | 0.755 | -0.012 |
| 20% | 69.6 | -2.0 | 0.738 | -0.029 |
| 30% | 67.9 | -3.7 | 0.712 | -0.055 |
| 50% | 63.2 | -8.4 | 0.661 | -0.106 |

At 10% noise, classification accuracy drops by only about 0.8%, with IoU decreasing by 0.012; at 20% noise, accuracy decreases by approximately 2.0%. This demonstrates GECL's strong tolerance for minor to moderate LLM-generated errors. As noise increases, the top-1 weight of correct attributes in DM gradually decreases (from 0.42 to 0.26), but significant performance degradation only occurs at high noise levels ($\geq$ 30%). This demonstrates that DM's global distribution automatically "suppresses weights" for irrelevant attributes, thereby limiting the influence of erroneous attributes on final decisions. IoU shows degradation earlier than accuracy (e.g., IoU drops by 0.055 at 30% noise), meaning that even if the model retains some classification capability, its "explainability quality" declines before accuracy—consistent with the intuition that "erroneous attributes may still lead to imprecise evidence pointing.

**Human Evaluation of the quality of attributes.** To evaluate the quality of LLM-generated attributes, we introduced a manual evaluation method. To ensure the reliability of the knowledge base, we randomly select 600 attributes (approximately 19% of the database) and have five experienced annotators independently evaluate them. Specifically, we evaluate each attribute across four aspects. **Semantic correctness** measures whether the attribute description

Table 8: Human evaluation on CUB200.

| Quality Metric | Score (%) |
|---|---|
| Semantic Correctness | 90.8 |
| Logical Consistency | 94.4 |
| Scenario Applicability | 92.2 |
| Action Clarity | 85.7 |

aligns with factual accuracy. **Logical consistency** assesses whether relationships between attributes are clear and conflict-free. **Scenario applicability** evaluates whether the scenario specified by the attribute is consistent with the content(foreground/background). **Action clarity** determines whether actions are clearly identifiable and consistently defined. All four metrics use a binary scoring (1/0), so each rule receives five independent scores for each metric. The final percentage score is the average of 3000 scores from five annotators for the 600 attributes. This scoring method ensures that the final score is a consensus among all annotators, rather than the subjective result of any single annotator. As shown in Table. 8, manual evaluation results indicate that most attributes are precisely defined and semantically sound.

Table 9: Ablation of Parameter Efficiency

| Configuration | Params | ACC | PD | IoU | Inference Time (s/img) |
|---|---|---|---|---|---|
| GECL – Adapter 2 layers × 256 dim | 5.7 M | 72.49 | 5.13 | 0.767 | 0.19 |
| Smaller Adapter – 1 layer × 128 dim | 3.1 M | 71.82 | 5.41 | 0.762 | 0.17 |
| Larger Adapter – 3 layers × 512 dim | 11.9 M | 72.68 | 4.96 | 0.771 | 0.24 |
| Frozen Backbone (no adapter) | 0 M | 63.10 | 13.44 | 0.522 | 0.17 |

**Ablation of Parameter Efficiency.** Table. 9 reports an efficiency ablation evaluating the effect of trainable parameter capacity on continual performance. Freezing the backbone without an adapter leads to severe forgetting (PD to 13.44), indicating that eliminating parameter updates is not sufficient for stable continual learning. In contrast, both small and large adapters perform similarly to the default configuration, demonstrating that GECL does not rely on large trainable modules. Overall, GECL achieves the best trade-off between accuracy, interpretability, and efficiency with only 5.7 M trainable parameters.

**Ablation of the Optimal Distribution Module for Catastrophic Forgetting** To better illustrate DM's role in alleviating catastrophic forgetting, we conducted an additional adapter-only baseline, where the entire DM module is removed and the prediction is made purely via global CLIP-style similarity between the image embedding and a text prototype. Importantly, in this setting the LLM-generated attributes cannot contribute to the decision process, since no region–attribute alignment is performed. This ablation therefore isolates the influence of the OT module from the architectural effects of the lightweight adapter.

Results on CUB-200-2011 and mini-ImageNet are shown in Table. 11. On the fine-grained CUB dataset, removing DM leads to a substantial performance drop (-14%), consistent with the fact that discriminative evidence in CUB is concentrated in part-level cues (e.g., head color, wing pattern), which the DM module explicitly aligns with fine-grained textual attributes. In contrast, on the coarse-grained mini-ImageNet benchmark, global features already provide strong separability between categories, and the OT module mainly serves as a robustness enhancer in difficult samples (e.g., small objects, occluded instances), leading to a smaller accuracy reduction (-3%). These find-

Table 11: Ablation of the Optimal Distribution Module for Catastrophic Forgetting

| Dataset | GECL (with DM) | Adapter-only | Acc Drop | PD (with DM) | PD (No DM) |
|---------|----------------|--------------|----------|--------------|------------|
| CUB-200 | 72.49% | 58.4% ± 2.5% | -14.1% | 5.2 | 11.8 |
| mini-ImageNet | 84.90% | 81.2% ± 1.5% | -3.7% | 3.9 | 6.1 |

ings validate that the DM reasoning mechanism provides complementary discriminative power that cannot be recovered by a parameter-efficient adapter alone, especially in fine-grained incremental settings. Additionally, it is important to note that DM not only plays a role in the classification process but is also an indispensable component of the interpretability process. Removing DM would result in the model completely losing its interpretability. PD represents performance drop(First stage accuracy−Final stage accuracy).

**Ablation of the LLMs choose.** GECL indeed treats the LLM as an offline semantic generator, and its design does not rely on any specific language model architecture. As we've reported in sec. 4.2, we used MiniGPT-4 (with the publicly released projection layer checkpoint) because it can provides stable part-level descriptions, and it is widely available for academic comparison.

To address this sensitivity, we have conducted an additional small-scale ablation in which we regenerate attributes using another Qwen2.5-VL-7B. The results in Table. 10 indicate that GECL is not sensitive

Table 10: Ablation of the LLMs choose

| LLM (attribute generator) | Final Acc (%) | PD (%) | IoU |
|---------------------------|---------------|--------|-----|
| MiniGPT-4 (Vicuna 7B) | 72.49 | 23.55 | 0.767 |
| Qwen2.5-VL-7B | 73.12 | 24.1 | 0.771 |

to the specific LLM backbone, as long as the generated attributes remain syntactically valid and roughly descriptive. This is consistent with the fact that GECL relies on distributional alignment rather than any LLM-specific embedding space or latent knowledge.

# E RELATED WORK

## E.1 CLASS-INCREMENTAL LEARNING

In class-incremental learning (CIL), tasks arrive sequentially and each class belongs exclusively to one task, with the goal of acquiring new classes while retaining previously learned ones. Existing approaches can be broadly categorized into rehearsal-based, architecture-based, and regularization-based methods.

**Rehearsal-based strategies** store exemplars from old classes to replay during training, which can be raw data (Rebuffi et al., 2017a), generative data (Gao & Liu, 2023; Shin et al., 2017), or hidden features (Hayes et al., 2020). **Architecture-based approaches** dynamically expand or reconfigure network parameters to mitigate forgetting, such as learning redundant structures (Fernando et al., 2017), expert networks (Aljundi et al., 2017; Schwarz et al., 2018), or task-specific parameters (Liu et al., 2021; Mallya et al., 2018; Serra et al., 2018). **Regularization-based methods** add constraints to prevent large updates to important parameters, e.g., EWC (Kirkpatrick et al., 2017a), SDC (Yu et al., 2020) , and Rotated-EWC (Liu et al., 2018), or enforce output consistency via distillation (Hu et al., 2021; Li & Hoiem, 2018; Liu et al., 2022b; Tao et al., 2020a; Zhang et al., 2022). In addition, other works classify incremental methods into regularization (Aljundi et al., 2018; Kirkpatrick et al., 2017a), replay (Liu et al., 2020; Lopez-Paz & Ranzato, 2017; Luo et al., 2023; Sun et al., 2023) , and dynamic architecture strategies (Douillard et al., 2022a; Wang et al., 2022a; Wu et al., 2022; Yan et al., 2021). Bias-correction strategies have also been proposed to mitigate classifier bias toward recent classes, such as CWR (Lomonaco & Maltoni, 2017), CWR+ (Maltoni & Lomonaco, 2019), and AR1 (Zeno et al., 2021).

Few-shot class-incremental learning (FSCIL) (Mazumder et al., 2021; Tao et al., 2020b) considers the setting where the base session provides full data and incremental sessions only contain very limited samples. Some methods train across both base and incremental sessions to reduce overfitting (Cheraghian et al., 2021; Tao et al., 2020b; Zhao et al., 2024), while others primarily rely on

the base session and apply minimal adaptation during incremental sessions (SHI et al., 2021; Zhang et al., 2021a; Zhu et al., 2021b).

## E.2 PRE-TRAINED MODELS AND PROMPTING FOR CIL

Pre-trained models have recently shown strong potential for continual learning (Thengane et al., 2022a; Wang et al., 2022d; Wu et al., 2022). One direction fine-tunes the pre-trained backbone with different learning rates (Zhang et al., 2023) or by distillation with external data (Zheng et al., 2023). Another direction freezes the backbone while training additional parameters such as adapters (Liu et al., 2023) or prompts (Khan et al., 2023; Smith et al., 2023a; Wang et al., 2022b;d). In particular, Continual-CLIP (Thengane et al., 2022a) demonstrates that CLIP (Radford et al., 2021) can perform continual learning without extra training. Other works like SLCA (Zhang et al., 2023) fine-tune CLIP for lower forgetting, while methods such as PROOF (Zhou et al., 2025) and RanPac (McDonnell et al., 2023)introduce cross-modal attention or high-dimensional projections. Prompt-based strategies such as DualPrompt (Wang et al., 2022d)and CODA-Prompt (Wang et al., 2022b)retain past knowledge by selecting subsets of prompts. However, classification solely relying on expanded text features aggravates the bias toward current data, leading to forgetting of old knowledge.

## E.3 GENERATIVE METHODS AND GENERATIVE CLASSIFIERS

Replay-based approaches often rely on exemplars, but when storing old data is infeasible due to privacy or memory constraints, generative replay has been explored (Aljundi et al., 2017; 2019b; Gao & Liu, 2023; van de Ven & Tolias, 2019). While effective for low-dimensional tasks, generative replay struggles with complex visual inputs (van de Ven, 2020). Generative classifiers, in contrast, directly perform classification via generative modeling, avoiding reliance on discriminative heads. Recent studies (van de Ven et al., 2021) propose rehearsal-free generative classifiers based on energy models and VAEs (Kingma D P, 2013), showing strong resistance to catastrophic forgetting and task confusion. They can also operate in task-free settings (Aljundi et al., 2019a;b), support single-class learning (van de Ven et al., 2021), and require less computation compared to generative replay.

## F FUTURE DIRECTIONS

While GECL establishes a principled framework for distribution-level alignment between visual evidence and LLM-generated semantic attributes, several promising research directions remain open for exploration.

**Structured and adversarial robustness of semantic attributes.** The noise-injection experiments in the current version primarily evaluate resilience to random corruption. A natural extension is to study structured, biased, or adversarially perturbed attribute sets, e.g., attributes that are systematically misleading or semantically inconsistent. Such tests would more thoroughly stress the model's reliance on the alignment geometry, and may reveal additional opportunities for robustness-aware knowledge construction.

**Automated detection and correction of knowledge inconsistencies.** The present knowledge base is constructed in an offline, one-shot manner. Although the static-isolation design effectively prevents cross-class contamination, it does not automatically repair subtle conflicts or logical inconsistencies that may arise within a class's attribute set. Developing mechanisms for detecting incompatible attributes, refining them through cross-modal validation, or distilling more coherent attribute subsets over time constitutes an important direction for long-term, self-correcting systems.

**Continual expansion and maintenance of semantic knowledge.** Extending GECL to a setting where attribute distributions evolve across tasks, while ensuring stability, avoiding drift, and maintaining historical consistency, poses new challenges. This invites research into knowledge governance strategies such as continual attribute auditing, incremental semantic refinement, and principled criteria for merging, pruning, or re-weighting attributes as new tasks appear.

**Causal and intervention-based interpretability evaluations.** Beyond the current part-level IoU and human studies, future work may incorporate causal interpretability metrics, such as dele-

tion/insertion tests and intervention-driven probing. These tools can provide a more rigorous view of the faithfulness and necessity of the region–attribute alignments that GECL produces.

**Integrating visual-context–conditioned prompting.** As observed in the prompt ablation studies, the quality of LLM-generated attributes depends on the specificity of the prompt. A promising future direction is to incorporate visual-context–conditioned prompting, e.g., using coarse visual descriptors, few-shot exemplars, or pseudo-caption summarization to make attribute generation more robust and less sensitive to lexical cues.

Taken together, these directions aim to extend GECL from a one-shot, static knowledge-alignment system toward a fully continual, self-maintaining, and causally grounded framework for interpretable incremental learning.

