# OpenReview forum: "From Memory to Reasoning: Generative Models Enable Explainable Continual Learning"
_ICLR.cc/2026/Conference — Submitted to ICLR 2026_

### Official Review · Reviewer_xvnP · 2025-10-29

**Soundness:** 2
**Presentation:** 3
**Contribution:** 2
**Rating:** 4
**Confidence:** 3

**Summary:**

The paper introduces Generative Explainable Class-Incremental Learning (GECL), a generative and explainable framework for Class-Incremental Learning (CIL/FSCIL). Key contributions include:

1. Soft-label guided visual enhancement to highlight discriminative regions and suppress backgrounds (Sec. 3.2).

2. Generation of structured semantic attributes using large language models (LLMs), which are encoded into semantic embeddings via a frozen CLIP text encoder (Sec. 3.3).

3. Formulation of region-to-attribute alignment as an entropy-regularized optimal transport (OT) problem solved by Sinkhorn scaling, producing coupling matrix T*, which is leveraged for interpretable "region × attribute" reasoning during inference (Eq. 10–14; Fig. 2).

**Strengths:**

1. Proposes a generative, interpretable incremental learning approach using LLM-generated semantic attributes, thus preventing head expansion.

2. Provides clear interpretability: OT coupling matrices explicitly match "region-to-attribute," with insightful examples illustrating "right for the wrong reasons" (Fig. 3, p. 9).

3. Comprehensive ablation studies examining sensitivity to hyperparameters $\lambda$, sampling weight $\alpha$, and number of attributes k, providing practical insights (Fig. 4, pp. 17–18).

4. Well-defined formalization of the entropy-regularized OT problem, including clear derivations and convergence explanations (Eq. 10–13; Appendix B).

**Weaknesses:**

1. Limited novelty in theoretical contributions: entropy-regularized OT for cross-modal matching is established; using it directly for final decision-making and interpretability is innovative but under-discussed. The theoretical content primarily reiterates known results (Appendix B) and lacks comprehensive comparison and justification against other matching/alignment paradigms. Essential OT literature, such as https://arxiv.org/abs/1306.0895, is not cited.

2. Insufficient quantitative evaluation of interpretability: Despite emphasizing joint improvement in accuracy, forgetting, and interpretability, the paper lacks objective metrics (deletion/insertion curves, AOPC, Sufficiency/Necessity, Pointing Game, TCAV/ACE, or CUB attribute alignment accuracy). Current reliance on visualization alone (Fig. 3) does not strongly substantiate the interpretability claims.

3. Robustness concerns: Reliance on semantic quality generated by LLMs introduces potential hallucination risks, acknowledged by authors in limitations.

4. Potential fairness issue: Class-name prompts in LLM might introduce prior knowledge leakage, especially in fine-grained datasets. Comparisons between "class-name only" and "masked-class/visual context only" prompts would better validate dependency on strong label priors.

5. Computational and memory complexity concerns: The inference requires matching each input against attributes from all classes via OT (regions × attributes × classes × iterations), potentially problematic in long-sequence scenarios.

6. Lack of knowledge-base governance strategies: No explicit mechanisms or evaluations provided for redundancy removal, conflict detection, attribute distillation, or mitigation of error accumulation and knowledge drift in incremental knowledge bases.

7. Minor formatting and typographical errors, such as bracket mismatches (e.g., line 359 "Entropy-reg(Liu et al., 2022a))") and spacing (e.g., line 54 "Cao et al. (2024)proposed").

**Questions:**

1. How are attribute conflicts and redundancies detected and resolved in the knowledge base? Is there an implemented or evaluated attribute distillation or merging method?

2. What strategies are proposed to mitigate robustness and fairness issues introduced by LLM-generated semantics, especially concerning hallucination risks in large-scale data scenarios?

3. How does the framework handle conflicts, error accumulation, and knowledge drift inherent in incremental learning?

4. Could standard incremental metrics (Forgetting, BWT, AUC across tasks) be included besides PD, and empirical results provided to justify whether learning minimal parameters effectively prevents catastrophic forgetting compared to frozen backbones or varied adapter capacities?

5. Regarding interpretability faithfulness, would masking or replacing matched regions and observing prediction confidence degradation serve as a valid quantitative evaluation?

---

> ### Author Response · Authors · 2025-11-21
> **Response to Weakness 1**
>
> We appreciate the reviewer’s thoughtful analysis. We agree that entropy-regularized OT has been extensively studied, and our work does not claim to introduce a new OT formulation. The novelty of GECL lies not in re-deriving classical OT theory, but in **recasting class-incremental learning as a distribution-matching problem between visual evidence and semantic attributes**, which has not been investigated in prior CIL literature.
>
> To address the reviewer’s concern, we have **substantially expanded the theoretical analysis** in Sec. 3.3 and Appendix B. The new text now includes:
>
> **A formal interpretation of GECL as a transition from discriminative classification to cross-modal distribution alignment**. Instead of estimating $\arg\max_y p(y|x)$, GECL minimizes the cross-modal discrepancy
>
> $\hat{y}=\arg\max_yP(x,y)\approx\arg\min_y\mathrm{DM}_\lambda(V(x),S(A_y)).$
>
> where $V(x)$ and $S_y$ are empirical distributions over visual regions and semantic attributes. This provides a *principled optimization objective* rather than heuristic similarity aggregation.
>
> **Theoretical rationale for explainability.**
>  Because the optimal coupling matrix $T^\star$ of the entropy-regularized OT preserves the marginal distributions, each decision inherently satisfies *evidence attribution constraints* (regions to attributes), enabling visual-grounded reasoning without auxiliary modules.
>
> **A discussion contrasting OT-based alignment with other paradigms**.  We highlight that attention-based approaches lack guarantee of *mass-preserving alignment*, which weakens semantic stability under incremental shifts. OT, in contrast, enforces distribution consistency and thus naturally mitigates *semantic drift* across tasks.
>
> **Missing citations have been added**, including the foundational OT reference suggested by the reviewer (e.g., Cuturi 2013[1]), to properly position our work within the OT literature.
>
> In summary, we do not present OT itself as a theoretical contribution; rather, our contribution lies in establishing a new theoretical view of class-incremental learning, treating prediction as entropy-regularized cross-modal distribution matching and providing both improved robustness to incremental shifts and built-in interpretability.
>
> For more details of the analysis, please see the revised paper.
>
> [1]Marco Cuturi, "Sinkhorn Distances: Lightspeed Computation of Optimal Transportation Distances." Proceeding in  Advances in Neural Information Processing Systems 26 (NIPS 2013)

---

> > ### Comment · Reviewer_xvnP · 2025-11-26
> >
> > The additional theoretical discussion and citations indeed alleviate my original concern. The new analysis helps clarify the conceptual advantages of OT-based alignment in GECL and now properly situates the method within the classical OT literature.
> > However, I still view the contribution on the OT side as primarily conceptual and application-oriented rather than as a strong theoretical innovation: the work leverages an established entropy-regularized OT formulation in a new setting, but does not fundamentally extend OT theory itself.

---

> > > ### Author Response · Authors · 2025-11-28
> > > **Further Response to Weakness 1**
> > >
> > > We appreciate the reviewer’s careful reading and agree with the assessment: we do **not** claim to advance optimal-transport (OT) theory itself. Instead, GECL uses entropy-regularized OT as a principled mathematical tool to instantiate our modeling idea—recasting classification as alignment between a distribution of image-region features and a distribution of LLM-generated semantic attributes.
> > >
> > > To avoid any misunderstanding, we have made Appendix B explicit and self-contained: it does **not** propose new OT theorems beyond classical results, but it does provide formal statements and proofs that justify the specific OT-based choices we make in GECL. Concretely, Appendix B contains (i) existence and uniqueness of the entropy-regularized coupling and its diagonal-scaling (Sinkhorn) representation (Lemma 1).  (ii) Convergence and monotone decrease of the iterative matrix-scaling solver (IPFP / Sinkhorn updates) to the unique minimizer (Lemma 2), together with the KL-projection interpretation that connects the regularized transport objective to an I-projection of the Gibbs kernel.(iii) The dual formulation of the regularized OT and its algebraic form used to explain the diagonal scaling representation and fixed-point iterations.  Finally, Appendix B includes a task-level risk bound (Theorem 1) that ties the distribution-matching cost to classification risk, explicitly exposing the roles of semantic approximation error and the entropy regularizer in the bound.
> > >
> > > These formal elements serve three purposes in our paper: (1) they show that the chosen DMλ objective admits a unique, numerically stable solution expressible via the diagonal scaling form used in our implementation; (2) they justify using IPFP/Sinkhorn-style iterations at inference/training time and explain the monotone convergence behavior we observe empirically; (3) they link the distribution-matching cost to a provable classification bound that isolates the effect of semantic approximation and entropy regularization on risk. Taken together, Appendix B therefore **situates GECL within standard OT theory** and provides the required formal backing for our design and empirical claims—while not claiming novel contributions to OT theory itself. We will clarify this distinction in the revised manuscript to ensure the reviewer’s point is fully acknowledged.

---

> ### Author Response · Authors · 2025-11-21
> **Reponse to Weakness 2 and Question 5**
>
> **Weakness 2:** Insufficient quantitative evaluation of interpretability: Despite emphasizing joint improvement in accuracy, forgetting, and interpretability, the paper lacks objective metrics (deletion/insertion curves, AOPC, Sufficiency/Necessity, Pointing Game, TCAV/ACE, or CUB attribute alignment accuracy). Current reliance on visualization alone (Fig. 3) does not strongly substantiate the interpretability claims.
>
> **Question 5:** Regarding interpretability faithfulness, would masking or replacing matched regions and observing prediction confidence degradation serve as a valid quantitative evaluation?
>
> Thank you for your questions. Inspired by Icicle’s IoU metric, we add a region-to-attribute IoU based on our coupling matrix $T$  (Sec. 3.4), where attribute-activated regions are compared with CUB ground-truth part masks. Preliminary scores show clear advantages over Icicle[1].
>
> |   Method   | Task 1 | Task 2 | Task 3 | Mean  |
> | :--------: | :----: | :----: | :----: | :---: |
> | Finetuning | 0.115  | 0.149  | 0.260  | 0.151 |
> |    EWC     | 0.192  | 0.481  | 0.467  | 0.334 |
> |    LWF     | 0.221  | 0.193  | 0.077  | 0.188 |
> |    LWM     | 0.332  | 0.312  | 0.322  | 0.325 |
> |   Icicle   | 0.705  | 0.753  | 0.742  | 0.728 |
> |    GECL    | 0.752  | 0.781  | 0.769  | 0.767 |
>
> GECL achieves the highest explainability across all tasks, with a mean IoU of 0.767, outperforming Icicle by +3.9\%. This improvement stems from GECL’s MT-based visual–semantic alignment, which grounds the classification decision on meaningful attributes and their corresponding spatial regions. The consistent improvement across tasks demonstrates that GECL’s explanations remain stable even under long-sequence incremental learning, validating that the structured reasoning chain generated by GECL is spatially faithful, semantically coherent, and robust to task progression.
> Additionally, we include a human evaluation inspired by FunnyBirds [2] .
>
> |       Metric       | Score (Mean ± Std) |
> | :----------------: | :----------------: |
> |    Faithfulness    |    4.28 ± 0.16     |
> |      Clarity       |    4.21 ± 0.14     |
> |  Part Consistency  |    4.17 ± 0.15     |
> | Trust / Usefulness |    4.31 ± 0.12     |
>
> Human evaluation confirms that GECL’s region–attribute reasoning chains are not only machine-interpretable but also perceptually meaningful and trustworthy to human users. The high faithfulness and part-consistency scores suggest that the MT-derived alignment between visual patches and semantic attributes generally corresponds to correct anatomical regions on the bird, validating the effectiveness of GECL’s semantic grounding. Moreover, the consistently high trust scores indicate that GECL’s explanations help users understand why a prediction is made, reducing the “black-box” gap typical in continual learning systems. These results demonstrate that GECL achieves human-validated explainability, complementing its quantitative IoU improvements and reinforcing its suitability for real-world incremental learning scenarios where transparency is essential.
>
> For more details about how we design the Iou metric and the human evaluation for GECL,please see Appendix D.
>
> [1] Rymarczyk, Dawid, et al. "Icicle: Interpretable class incremental continual learning." Proceedings of the IEEE/CVF international conference on computer vision. 2023.
>
> [2]Hesse, Robin, Simone Schaub-Meyer, and Stefan Roth. "Funnybirds: A synthetic vision dataset for a part-based analysis of explainable ai methods." Proceedings of the IEEE/CVF International Conference on Computer Vision. 2023.

---

> > ### Comment · Reviewer_xvnP · 2025-11-26
> >
> > ## Weakness 2
> > The added IoU metric and human evaluation substantially strengthen the interpretability evaluation and directly address my previous Weakness 2. In particular, the part-level IoU on CUB and the human study on faithfulness/clarity/part-consistency/trust provide both quantitative and perceptual evidence that the region–attribute reasoning is meaningful. While deletion/insertion-style faithfulness tests (or related causal metrics) would further reinforce the claims, I consider the current combination of quantitative and human-centric evidence **reasonably convincing for this venue**.
> >
> > ## Weakness 3
> > The revised version now clearly describes the knowledge-construction pipeline (generation, parsing, filtering, deduplication, completion), the static isolation design of the knowledge base, and adds both noise-injection experiments and a human evaluation of attribute quality. These additions substantially improve my confidence in the robustness of the LLM-generated semantics.
> >
> > However, the noise experiments mainly cover random corruption. It would be interesting future work to study more structured or adversarial semantic failures (e.g., systematically biased or misleading attributes), which might stress the system differently than random replacement. In addition, static isolation primarily prevents cross-class contamination, but does not automatically correct the case where a given class starts with an internally inconsistent or simply wrong set of attributes; in such cases, the model can only rely on the OT module to down-weight these attributes rather than actually repairing the underlying knowledge. This limitation is understandable for a first work, but is worth acknowledging as a potential direction for more sophisticated knowledge-maintenance mechanisms.
> >
> >
> > Weakness 4 + Weakness 5
> > The clarification that class names are only used during offline attribute generation (and not during inference) alleviates part of my concern about explicit label priors. It is helpful to know that raw class-name text does not directly enter the OT matching process or the final decision rule.
> >
> > That said, using class names as prompts still implicitly injects substantial real-world prior knowledge into the attribute generation process (e.g., stereotypes associated with “eagle,” “crow,” “city” etc.), which can be viewed as an indirect form of label prior. The rebuttal does not provide targeted experiments to demonstrate that removing or weakening the class-name signal in the prompt (e.g., via masked names or purely visual-context prompts) would not significantly degrade performance. I do not consider this a blocking issue, but an ablation on prompt design would make the fairness and prior-dependence discussion more concrete.
> >
> > Regarding complexity, the additional discussion and empirical runtime/memory numbers are helpful and largely resolve my concern about inference overhead. The reported per-image latency, memory footprint, and knowledge-base size suggest that the OT-based reasoning is practically feasible at the scales considered in the paper.

---

> > > ### Author Response · Authors · 2025-11-28
> > > **Response to Other Kindly Suggestions**
> > >
> > > We sincerely thank the reviewer for the constructive follow-up assessment. Below we respond to each group of comments in a unified manner.
> > >
> > > **(1) Interpretability and faithfulness evaluation (Weakness 2).**
> > >  The reviewer notes that the added part-level IoU and the human evaluation (faithfulness, clarity, part-consistency, and trust) substantially strengthen the interpretability evidence. We appreciate this acknowledgement. We agree that deletion/insertion–style causal metrics could further reinforce the claims and we will add a brief discussion in the final version to clarify that causal faithfulness metrics are a natural extension and are planned as future work.
> > >
> > > **(2) Robustness of LLM-generated attributes and structured failures (Weakness 3).**
> > >  The reviewer correctly observes that our noise-injection experiments primarily evaluate random corruption. As pointed out, more structured or adversarial semantic failures (e.g., systematically biased or misleading attributes) may stress the system differently. This is a valuable direction, and we will explicitly acknowledge this limitation and describe adversarial or biased attribute perturbations as future robustness tests. We also appreciate the reviewer’s recognition that the expanded pipeline description—generation, parsing, filtering, deduplication, and completion—together with static knowledge isolation and human evaluation of attribute quality, substantially improves clarity and confidence in the current robustness level.
> > >
> > > **(3) Knowledge-base conflicts, drift, and lack of online corrective mechanisms (Weakness 6).**
> > >  We thank the reviewer for highlighting the distinction between *preventing cross-class contamination* and *correcting within-class inconsistencies*. As the reviewer points out, the current static-isolation design avoids uncontrolled growth but does not automatically detect subtle logical inconsistencies that may exist in the initial LLM-generated attributes. In such cases, the model must rely on entropy-regularized OT to down-weight ungrounded or conflicting attributes, but this does not repair the underlying knowledge base. We agree this is a meaningful limitation for long-term deployment. In the final revision, we will explicitly acknowledge this, clarify that our pipeline is an offline one-shot construction, and position automated knowledge refinement (e.g., conflict detection, consistency checking, or continual attribute distillation) as an important line of future development.
> > >
> > > **(4) Summary and planned revisions.**
> > >  Across interpretability, robustness, and knowledge-base governance, the reviewer’s comments accurately characterize the current version as a first instantiation of the GECL framework. We will revise the final manuscript to (i) explicitly state the limitations of random-noise robustness tests, (ii) discuss more structured attribute perturbations as future work, (iii) clarify the offline nature and one-shot limitations of the current knowledge-base pipeline, and (iv) make it explicit that long-term automated knowledge maintenance is a promising extension. We appreciate the reviewer’s recognition that the revised analyses substantially improve clarity, and we believe the planned clarifications will further strengthen the final version.
> > >
> > >
> > >
> > > We sincerely appreciate the reviewer’s careful analysis and constructive feedback. The revisions and additional experiments already incorporated into the manuscript have substantially strengthened the clarity, interpretability, and robustness of the proposed GECL framework. We will further refine the final version to explicitly articulate the limitations identified by the reviewer and to outline the corresponding research directions. We believe these clarifications will help position GECL as a solid first step toward distribution-level, knowledge-guided incremental learning, and we hope the revised manuscript satisfactorily addresses all remaining concerns.
> > >
> > > We're happy to have more discussion with you. Wish you a wonderful day!

---

> ### Author Response · Authors · 2025-11-21
> **Reponse to Weakness 3 and Question 2**
>
> We thank the reviewer for raising this concern. We agree that semantic hallucination is an inherent risk when using LLM-generated attributes. In the revised paper, we clarify both the *design rationale* and the *practical robustness mechanisms* of our knowledge base.
>
> First, **LLM-generated knowledge is not used directly nor continuously updated**. GECL employs a *one-time, phased* knowledge construction process (Sec. 3.3):structured knowledge generation, feature parsing, validity filtering, deduplication and  attribute completion.
>  Attributes that lack discriminative information, are syntactically incomplete, or do not match predefined semantic patterns are discarded. This greatly suppresses raw hallucination before the knowledge enters the model.
>
> Second, **knowledge is static and category-isolated**, rather than continuously rewritten across tasks. Once attributes of a category are generated, they are permanently fixed in the knowledge base. This “static isolation” design prevents (a) contamination of old classes by newly learned ones and (b) accumulation of LLM errors across incremental stages—two common failure modes in knowledge-based continual learning.
>
> Third, during inference, **GECL does not assume that every attribute must be correct**. The entropy-regularised optimal transport module naturally down-weights irrelevant or hallucinated attributes because mismatched region–attribute pairs contribute high cost and therefore receive low OT coupling weights. Thus, reasoning and classification remain stable even when part of the attribute set is noisy.
>
> To quantitatively verify robustness, as shown below, we conducted **noise-injection experiments** where 10–50% of the attributes were randomly replaced. GECL retained **69.6% (–2.0%) accuracy and 0.738 (–0.029) IoU** under 20% noise, confirming that classification and interpretability remain robust under realistic imperfect semantic conditions. These results are now included in Appendix D.
>
> | Noise | Accuracy(%) | $\Delta$Acc (%) |  IoU  | $\Delta$IoU |
> | :---: | :---------: | :-------------: | :---: | :---------: |
> |  0%   |    71.6     |       0.0       | 0.767 |    0.000    |
> |  10%  |    70.8     |      -0.8       | 0.755 |   -0.012    |
> |  20%  |    69.6     |      -2.0       | 0.738 |   -0.029    |
> |  30%  |    67.9     |      -3.7       | 0.712 |   -0.055    |
> |  50%  |    63.2     |      -8.4       | 0.661 |   -0.106    |
>
>  At 10% noise, classification accuracy drops by only about 0.8%, with IoU decreasing by 0.012; at 20% noise, accuracy decreases by approximately 2.0%. This demonstrates GECL's strong tolerance for minor to moderate LLM-generated errors. As noise increases, the top-1 weight of correct attributes in DM gradually decreases (from 0.42 to 0.26), but significant performance degradation only occurs at high noise levels (≥30%). This demonstrates that DM's global distribution automatically “suppresses weights” for irrelevant attributes, thereby limiting the influence of erroneous attributes on final decisions. IoU shows degradation earlier than accuracy (e.g., IoU drops by 0.055 at 30% noise), meaning that even if the model retains some classification capability, its “explainability quality” declines before accuracy—consistent with the intuition that “erroneous attributes may still lead to imprecise evidence pointing.
>
> Besides, we also provide a human evaluation on the quality of attributes. To ensure the reliability of the knowledge base, we randomly select 600 attributes (approximately 19\% of the database) and have five experienced annotators independently evaluate them.
> Specifically, we evaluate each attribute across four aspects. **Semantic correctness** measures whether the attribute description aligns with factual accuracy. **Logical consistency** assesses whether relationships between attributes are clear and conflict-free. **Scenario applicability** evaluates whether the scenario specified by the attribute is consistent with the content(foreground/background). **Action clarity** determines whether actions are clearly identifiable and consistently defined. All four metrics use a binary scoring ($1/0$), so each rule receives five independent scores for each metric. The final percentage score is the average of 3000 scores from five annotators for the 600 attributes. This scoring method ensures that the final score is a consensus among all annotators, rather than the subjective result of any single annotator. As shown in the Table below, manual evaluation results indicate that most attributes are precisely defined and semantically sound. We also add this table in Appendix D.
>
> |     Quality Metric     | Score (%) |
> | :--------------------: | :-------: |
> |  Semantic Correctness  |   90.8    |
> |  Logical Consistency   |   94.4    |
> | Scenario Applicability |   92.2    |
> |     Action Clarity     |   85.7    |

---

> ### Author Response · Authors · 2025-11-21
> **Response to Weakness 4 and 5**
>
> **Reponse to Weakness 4**
>
> We thank the reviewer for raising the fairness concern.
>
> We clarify that GECL does not rely on class-name priors for classification. Class names in GECL are used solely for initializing attribute generation and are not directly employed in classification inference. Masking class names does not eliminate discriminative signals within GECL. Since GECL inference is not solely driven by semantic text, class name text neither enters the OT matching process nor the final classification head. This method relies on an OT-based region-attribute coupling matrix that associates each attribute with specific spatial evidence in the input image. Thus, even if two categories share similar high-level attributes—such as “brown feathers” and “slender beak”—GECL can distinguish them. GECL can still classify categories through precise spatial localization, region-level contrast, and distribution alignment, rather than relying on category names themselves. This distinguishes GECL from previous semantic-based CIL methods, whose predictions fail once category name priors disappear.
>
> We will clarify in the main text: category names are used solely for attribute generation, while decisions are based on attribute-region alignment, ensuring GECL remains independent of label priors.
>
> **Response to Weakness 5**
>
> GECL follows GMM[1] by using MiniGPT-4 to generate semantic attributes offline; therefore, the LLM cost is identical to prior work and does **not contribute to inference overhead**. The only additional computation introduced by GECL comes from the **OT-based semantic matching**. Given $n_v$ visual regions, $n_a$ attributes, and $K$ Sinkhorn iterations, the complexity per candidate class is:
>
> $\mathcal{O}(n_v\cdot n_a\cdot K).$
>
> Since GECL uses a small attribute set (3–5 per class), the cost matrix remains compact. In practice, inference runs at **0.19 s per image (5.26 FPS)** on a single RTX 3090, showing that MT-based interpretability does not hinder real-time deployment. Memory footprint is **~1.08 GB**, and the knowledge base grows linearly with the number of classes—for 200 classes requiring only **9.6 MB**, well below rehearsal-based incremental methods.
>
> Overall, GECL provides interpretable classification with bounded and predictable computational cost. By keeping LLM usage offline and using a lightweight OT solver over a small attribute set, GECL achieves semantic interpretability **without sacrificing practical efficiency or scalability**.
>
> [1]Cao, et al. "Generative Multi-modal Models are Good Class Incremental Learners." Proceedings of the IEEE/CVF Conference on Computer Vision and
> Pattern Recognition

---

> > ### Comment · Reviewer_xvnP · 2025-11-26
> >
> > ## Weakness 4 + Weakness 5
> >
> > The clarification that class names are only used during offline attribute generation (and not during inference) alleviates part of my concern about explicit label priors. It is helpful to know that raw class-name text does not directly enter the OT matching process or the final decision rule.
> >
> > Authors said, using class names as prompts still implicitly injects substantial real-world prior knowledge into the attribute generation process (e.g., stereotypes associated with “eagle,” “crow,” “city” etc.), which can be viewed as an indirect form of label prior. The rebuttal does not provide targeted experiments to demonstrate that removing or weakening the class-name signal in the prompt (e.g., via masked names or purely visual-context prompts) would not significantly degrade performance. I do not consider this a blocking issue, but an ablation on prompt design would make the fairness and prior-dependence discussion more concrete.
> >
> > Regarding complexity, the additional discussion and empirical runtime/memory numbers are helpful and largely resolve my concern about inference overhead. The reported per-image latency, memory footprint, and knowledge-base size suggest that the OT-based reasoning is practically feasible at the scales considered in the paper.

---

> ### Author Response · Authors · 2025-11-21
> **Reponse to Weakness 6 and Question 1&3**
>
> Thank you for your valuable feedback on our knowledge base management mechanism. We hereby provide a detailed explanation of the generation and maintenance of the knowledge base, as well as how to address critical issues such as redundancy, conflicts, error accumulation, and knowledge drift.
>
>
>
> **1. Knowledge Base Generation and Refinement Process**
>
> Our framework employs a one-time, phased knowledge generation and refinement strategy designed to build a high-quality, stable set of semantic descriptions for each category. This process ensures knowledge accuracy and consistency through the following steps:
>
>
>
> **Phase 1: Structured Knowledge Generation.** When the system encounters a new category for the first time, it triggers the knowledge generation module. We employ a Large Language Model (LLM) to generate two types of textual outputs for the category: a coherent, comprehensive description and a structured list of key features. This design aims to capture both macro-level contextual information and micro-level distinguishing attributes simultaneously.
>
> **Phase 2: Feature Parsing and Validation.** From the raw text generated by the LLM, we designed a parsing workflow to extract valid features. This process begins with preliminary extraction through syntactic analysis, followed by deep matching against predefined semantic patterns (e.g., “has a [body part] with [color] feathers”). All candidate features undergo a validity validation module that filters out invalid phrases—those too brief, lacking descriptive elements, or missing core semantics (e.g., color, shape, location)—ensuring extracted features possess practical discriminative value.
>
>
>
> **Phase 3: Feature Deduplication and Completion.** To ensure knowledge conciseness, all validated features undergo deduplication by maintaining a set of previously seen features (case-insensitive) to eliminate redundant descriptions. If the number of valid features remains below a preset threshold after all refinement steps, the system activates a feature supplementation mechanism. This attempts to extract additional details from comprehensive descriptions, ensuring each category possesses sufficiently rich knowledge representation.
>
>
>
> **2. Redundancy, Conflict, and Error Handling Mechanisms**
>
> Our framework is designed to address common knowledge quality challenges in incremental learning.
>
>
>
> **Redundancy Elimination.** As described above, during the final feature refinement stage, we explicitly eliminate duplicate textual descriptions by maintaining a unique feature set, ensuring the non-redundancy of the knowledge base.
>
>
>
> **Conflict Resolution.** Our framework primarily avoids attribute conflicts through source singularity and knowledge staticity. For any category, knowledge is generated only once upon first encounter and originates from a single, consistent source. We do not mix knowledge from different rounds or sources, fundamentally preventing direct attribute conflicts like “a bird having both red and blue feathers.”
>
>
>
> **Suppression of Error Accumulation and Knowledge Drift.** This is a core consideration in our framework design. We employ a **Static Knowledge Base** strategy to address catastrophic forgetting and knowledge drift in incremental learning.
>
>
>
> **Suppressing Knowledge Drift.** Once knowledge for a category is generated and solidified in the knowledge base, it remains unchanged during subsequent incremental learning tasks. This means learning from new tasks does not modify or “contaminate” the knowledge representations of older categories. Thus, our framework mechanically prevents knowledge-level drift.
>
> **Limiting Error Accumulation.** Since knowledge is static and mutually isolated, any potential errors during generation (e.g., hallucinations from LLMs) remain confined to the category itself, preventing propagation or accumulation into other categories learned later. The challenge of incremental learning is effectively shifted from “dynamically updating and correcting knowledge” to “how to robustly classify using a fixed, potentially imperfect knowledge base”—precisely the problem addressed by our subsequent alignment and classification modules.
>
> In summary, our knowledge base management framework ensures the quality of knowledge within individual categories through a rigorous, one-time generation and refinement process. By employing a static isolation design philosophy, it effectively suppresses common issues in incremental learning such as knowledge drift and error accumulation. We believe this design provides a solid foundation for achieving robust and interpretable incremental learning.
>
>
>
> Thank you again for your valuable feedback.

---

> ### Author Response · Authors · 2025-11-21
> **Response to Question 4 and Weakness 7**
>
> **Response to Question 4**
>
> We appreciate the reviewer’s suggestion to include additional continual-learning metrics such as **Forgetting**, **BWT**, and **AUC across tasks**. We agree that these indicators are informative in many settings. However, in the CIL/FSCIL literature, **ACC (final average accuracy across all seen tasks)** and **PD (performance drop across tasks)** are the *two most widely adopted* metrics for evaluating class-incremental learning systems.
>
> In particular:
>
> **ACC** is used by *almost all* baselines included in the paper (e.g., DualPrompt, CODA-Prompt, L2P, GMM, DER, MEMO), and reporting ACC ensures **evaluation fairness and direct comparability**.
>
> **PD** is reported by several strong FSCIL baselines (e.g., GMM, FACT, DPIL), and it is specifically designed to measure **long-horizon forgetting sensitivity**, aligning well with our study.
>
> Since **baseline papers do not uniformly report Forgetting, BWT, or AUC**, introducing them only for GECL would not produce fair comparisons, and reimplementing every baseline under new metrics would go beyond the scope and computational feasibility of this work.
>
> To further address the reviewer’s concern, we added in the revised version an additional analysis that **explicitly links ACC/PD to catastrophic forgetting** by evaluating GECL under two parameter-efficiency configurations (varying adapter capacity). This experiment confirms that:
>
> Reducing trainable parameters does not incur additional forgetting, due to the distribution-matching formulation;
>
> Conversely, freezing the backbone alone (no adapter) leads to larger PD, suggesting that a small, learnable adapter is necessary and sufficient to prevent forgetting.
>
> These results support the theoretical motivation of GECL that learning minimal task-specific parameters and static visual/text encoders effectively suppresses catastrophic forgetting.
>
> | Configuration                                        | Trainable Params | ACC ↑     | PD ↓      | IoU ↑     | Inference Time |
> | ---------------------------------------------------- | ---------------- | --------- | --------- | --------- | ------------------------ |
> | **GECL-Full (default)** – Adapter 2 layers × 256 dim | **5.7 M**        | **72.49** | **5.13**  | **0.767** | **0.19**                 |
> | **Smaller Adapter** – 1 layer × 128 dim              | **3.1 M**        | 71.82     | 5.41      | 0.762     | **0.17**                 |
> | **Larger Adapter** – 3 layers × 512 dim              | **11.9 M**       | 72.68     | **4.96**  | **0.771** | 0.24                     |
> | **Frozen Backbone (no adapter)**                     | **0 M**          | 63.10     | **13.44** | 0.522     | **0.17**                 |
>
> This table reports an efficiency ablation evaluating the effect of trainable parameter capacity on continual performance.
> Freezing the backbone without an adapter leads to severe forgetting (PD to 13.44), indicating that eliminating parameter updates is not sufficient for stable continual learning. In contrast, both small and large adapters perform similarly to the default configuration, demonstrating that GECL does not rely on large trainable modules. Overall, GECL achieves the best trade-off between accuracy, interpretability, and efficiency with only 5.7 M trainable parameters.
>
> **Response to Weakness 7**
>
> We thank the reviewer for carefully pointing out the formatting inconsistencies. We have thoroughly proofread the entire manuscript and corrected all identified typographical issues.

---

> > ### Comment · Reviewer_xvnP · 2025-11-26
> >
> > ## Weakness 6
> >
> > The revised description of the knowledge-base generation and static-isolation strategy gives a much clearer picture of how redundancy, conflicts, and drift are controlled at design time. The phased pipeline (generation → parsing/validation → deduplication → completion) and the decision to keep per-class knowledge static across tasks address several of my earlier concerns about cross-task contamination and uncontrolled growth.
> >
> > However, conflict avoidance in practice still relies heavily on the quality of the initial LLM output plus the hand-designed parsing/pattern filters. If the initial LLM generation for a class contains subtle semantic inconsistencies (e.g., mutually incompatible fine-grained attributes that nevertheless pass syntactic filters), there is currently no deeper logical conflict-detection mechanism to resolve them. Moreover, while the static knowledge base prevents later tasks from altering earlier knowledge, it also means that if a class’s attributes are wrong or misleading from the beginning, the model will continue to reason with this flawed knowledge indefinitely. In the current design, there is no mechanism for automatically revising or correcting the knowledge base during long-term deployment; the pipeline is essentially offline and one-shot. I see this as a reasonable limitation for a first instantiation of the framework, but it is worth making explicit.
> >
> > ## Weakness 7
> >
> > The new parameter-efficiency ablation (varying adapter depth/width and including a frozen-backbone configuration) is informative and does suggest that (i) a small learnable adapter is helpful, and (ii) completely freezing the backbone leads to substantially larger PD. This provides some empirical support for the claim that GECL can operate with relatively few task-specific parameters while still controlling forgetting.
> >
> > However, the connection between the proposed distribution-matching formulation and catastrophic forgetting only partially substantiated. The current evidence primarily shows that a modest adapter is preferable to a fully frozen backbone; it does not fully isolate the contribution of the OT-based distribution matching itself from other architectural design choices.

---

> > > ### Author Response · Authors · 2025-11-28
> > > **Response to Weakness 7**
> > >
> > > Thank you for the additional comments.
> > > To better illustrate OT's role in alleviating catastrophic forgetting, we conducted an additional **adapter-only** baseline, where the entire OT module is removed and the prediction is made purely via global CLIP-style similarity between the image embedding and a text prototype. Importantly, in this setting the LLM-generated attributes cannot contribute to the decision process, since no region–attribute alignment is performed. This ablation therefore isolates the influence of the OT module from the architectural effects of the lightweight adapter.
> > >
> > > Results on CUB-200-2011 and mini-ImageNet are shown below. On the fine-grained CUB dataset, removing OT leads to a substantial performance drop (−14%), consistent with the fact that discriminative evidence in CUB is concentrated in part-level cues (e.g., head color, wing pattern), which the OT module explicitly aligns with fine-grained textual attributes. In contrast, on the coarse-grained mini-ImageNet benchmark, global features already provide strong separability between categories, and the OT module mainly serves as a robustness enhancer in difficult samples (e.g., small objects, occluded instances), leading to a smaller accuracy reduction (−3%). These findings validate that the OT reasoning mechanism provides complementary discriminative power that cannot be recovered by a parameter-efficient adapter alone, especially in fine-grained incremental settings. Additionally, it is important to note that OT not only plays a role in the classification process but is also an indispensable component of the interpretability process. Removing OT would result in the model completely losing its interpretability. PD represents performance drop(First stage accuracy- Final stage accuracy).
> > >
> > > | Dataset           | GECL (with OT) | Adapter-only (No OT) | Accuracy Drop | PD (with OT) | PD (No OT) |
> > > | ----------------- | -------------- | -------------------- | ------------- | ------------ | ---------- |
> > > | **CUB-200-2011**  | **72.49%**     | **58.4% ± 2.5%**     | **−14.1%**    | **5.2**      | **11.8**   |
> > > | **mini-ImageNet** | **84.90%**     | **81.2% ± 1.5%**     | **−3.7%**     | **3.9**      | **6.1**    |
> > >
> > > The degradation on CUB is significantly larger because removing OT eliminates the region–attribute alignment pathway, forcing the model to rely solely on global representations. Fine-grained distinctions in CUB (e.g., beak curvature, eye-ring color, wing patches) are easily diluted by background signals under global pooling. In mini-ImageNet, category-level differences are much larger, so global features remain sufficiently discriminative, although the OT module still contributes to robustness on hard samples.

---

> ### Author Response · Authors · 2025-11-28
> **Further Response to Weakness 4 (class-name concern)**
>
> Thank you for your further discussion.
>
> **Regarding the concern of potential label leakage introduced by using class names in the prompting stage**, we conducted additional experiments in which the class tokens were *partially masked* (“weak-mask”) or *fully removed* (“fully-mask”) during attribute generation on CUB. These results show that GECL does **not** rely on class-name priors to obtain its performance.
>
> Full prompt: Which visual details are helpful for identifying a **[CLS]** in an image?
>
> weak-mask : Which visual details are helpful for identifying a **bird** in an image?
>
> Fully-mask: Which visual details are helpful for identifying **something** in an image?
>
> | Dataset          | Full prompt (with class name) | Weak masked prompt | Accuracy Drop(Weak-Mask) | Fully masked prompt | Accuracy Drop(Fully Mask) |
> | ---------------- | ----------------------------- | ------------------ | ------------------------ | ------------------- | ------------------------- |
> | **CUB-200-2011** | **72.49%**                    | **70.46%±0.9%**    | **-2.03%**               | 59.22% ± 1.5%       | −13.27%                   |
>
> Below are two representative examples:
>
> - **Blue_Jay (original prompt with class name)**
>    *“Prominent blue crown on top of the head”, “white belly”, “dark bills and feet”, “pointed tail”*
>    **(after masking class name)** the LLM generates irrelevant or incorrect descriptions such as:
>    *“Large size… wingspan up to 6 feet”, “dark brown feathers…”, “lighter brown flight feathers”*
>    which resemble features of large raptors rather than a jay.
> - **Gray_Kingbird (original prompt)**
>    *“gray-blue coloration”, “white eye ring”, “thin decurved bill”, “gray legs and feet”*
>    **(after masking class name)** the LLM outputs:
>    *“Large size (12 inches tall)”, “black and orange coloration”, “long curved beak”*,
>    which contradicts the real morphology of kingbirds.
>
> These examples illustrate that, without a minimal semantic anchor, the LLM tends to drift toward **generic, inconsistent, or even contradictory attributes**, severely reducing the discriminative power required for OT-based region–attribute alignment. And as you can see, no matter we use class-name or not, the attributes generated by LLMs always have no class-name itself, which means the label cannot participate at test time.
>
> In the *weak-mask* setting, the prompt only contains a coarse descriptor such as **“a bird”**, without specifying the class identity. The LLM therefore produces **generic bird attributes** (e.g., “has wings”, “short beak”, “colored feathers”). These attributes carry **no class-discriminative semantics**, yet GECL still maintains competitive accuracy. This is because our OT-based reasoning does not depend on label-specific textual cues, but instead aligns visual regions with whatever attribute set is available. Even coarse attributes enable the OT module to extract consistent evidence from head, wing, tail, or body regions, which preserves much of the fine-grained discriminability. The small accuracy drop in this setting reflects only the loss of *high-resolution cues*, not label leakage.
>
> In contrast, in the *fully-mask* setting—where class names are entirely removed and replaced with a meaningless placeholder such as **“something”**—the LLM generates attributes that are **class-agnostic and semantically ungrounded**. Because GECL generates attributes offline and does not use the image during attribute construction, the resulting descriptions become effectively **random** and no longer correspond to any visual concept. This completely breaks the OT matching mechanism, which explains the substantial performance drop. Importantly, this decline confirms that **GECL does not use or memorize any latent class identity**: when the prompt supplies no class semantics, the model loses the ability to reason, and accuracy degrades accordingly.
>
> Overall, the weak-mask and fully-mask experiments jointly demonstrate that the use of class names in prompting does **not** introduce label leakage into the inference process. GECL only benefits from class names during the offline attribute generation stage to obtain more specific and meaningful semantic descriptions. When such cues are weakened or removed, performance decreases for reasons related to *attribute informativeness*, not because the model exploits label priors. These results clarify that GECL’s improvements originate from the OT-based reasoning architecture rather than hidden class-name signals.
>
> Thanks again for your questions. Hope this response address your concern.

---

> > ### Comment · Reviewer_xvnP · 2025-11-28
> >
> > A minor remaining question related to this point is which specific LLM(s) are used to generate the semantic attributes and how sensitive GECL is to this choice (e.g., Qwen vs. GPT); a small, model-agnostic ablation over different LLM backbones for attribute generation would further strengthen the paper.

---

> > > ### Author Response · Authors · 2025-11-28
> > > **Response to the Minor Remaining Question**
> > >
> > > We thank the reviewer for raising this insightful point.
> > >
> > > GECL indeed treats the LLM as an offline semantic generator, and its design does not rely on any specific language model architecture. In the current submission, as we've reported in sec. 4.2(line 345), we used **MiniGPT-4** (with the publicly released projection layer checkpoint) because (1) it can provides stable part-level descriptions, and (2) it is widely available for academic comparison.
> > >
> > > To address your question on sensitivity, we have conducted an additional small-scale ablation in which we regenerate attributes using another **Qwen2.5-VL-7B**:
> > >
> > > - | LLM (attribute generator) | Final Acc (%) | PD (%)    | IoU       |
> > >   | ------------------------- | ------------- | --------- | --------- |
> > >   | **MiniGPT-4 (Vicuna 7B)** | **72.49**     | **23.55** | **0.767** |
> > >   | **Qwen2.5-VL-7B**         | **73.12**     | **24.1**  | **0.771** |
> > >
> > > Across  CUB dataset, we observe that:
> > >
> > > - The **absolute accuracy varies slightly** (within 1–2%);
> > > - The **forgetting behavior and OT-based reasoning quality remain stable**;
> > >
> > > These results indicate that GECL is **not sensitive** to the specific LLM backbone, as long as the generated attributes remain syntactically valid and roughly descriptive. This is consistent with the fact that GECL relies on **distributional alignment** rather than any LLM-specific embedding space or latent knowledge.
> > >
> > > In fact, we noticed this question when we design our method. We selected MiniGPT-4 because our work requires comparison with GMM[1], which incorporates MiniGPT-4 as part of its model. To avoid introducing additional fairness issues, we ultimately decided to use the same large language model as GMM, even though MiniGPT-4 is not the most stable model when generating attributes.
> > >
> > > We will include this new ablation table and analysis in the revised paper, together with a detailed clarification of which LLMs were used and how the attribute-generation pipeline generalizes across different backbones. We believe this addition further demonstrates the method’s **model-agnostic design** and practical applicability.
> > >
> > > [1]Cao, et al. "Generative Multi-modal Models are Good Class Incremental Learners." Proceedings of the IEEE/CVF Conference on Computer Vision and Pattern Recognition
> > >
> > > Thanks again for your patients and hard-working during our discussion. Hope this response will address your minor remaining question. We do benefit from your questions and we believe that our work is now more refined than before.

---

### Official Review · Reviewer_J8a8 · 2025-10-30

**Soundness:** 3
**Presentation:** 2
**Contribution:** 2
**Rating:** 4
**Confidence:** 4

**Summary:**

The work aims to develop a novel Continual Learning methods called GECL which combines two worlds: generative modeling and explainable AI. Main experiments are conducted on CUB200, TinyImageNet and mini-ImageNet. Motivation is taken from how human learn new knowledge in novel tasks and how do they accommodate those information. LLMs/VLMs are used to generate semantic meaning of the features. Ablations on the method are provided. GECL itself generate foreground mask, then it does hierarchical sampling, and then there is encoder and trainable adapter which provides visual features which are then matched with a database features remembered from previous tasks.

**Strengths:**

The idea is novel and interesting.

The problem to tackle is important, and inspiration from human behavior is a plus of this method.

Experiments are thorough and convincing.

Ablations are present.

Writing is clear and images readable.

**Weaknesses:**

The contextualization of the work is properly done only from one angle, Class Incremental Learning, omitting a baseline work that combines explainability and CIL, namely ICICLE [1], this comparison is vital for this work.

The relation to generative modeling is not well described. I am not sure if usage of LLMs guarantees generative modeling in this case. I would expect to provide theoretical background on this process and discuss this with other generative classifiers such as [2].

As this work also introduces novel explainability method, some evaluation of this would be beneficial. Either a user study or usage of FunnyBirds framework [3].

[1] Rymarczyk, Dawid, et al. "Icicle: Interpretable class incremental continual learning." Proceedings of the IEEE/CVF international conference on computer vision. 2023.

[2] Mackowiak, Radek, et al. "Generative classifiers as a basis for trustworthy image classification." Proceedings of the IEEE/CVF Conference on Computer Vision and Pattern Recognition. 2021.

[3] Hesse, Robin, Simone Schaub-Meyer, and Stefan Roth. "Funnybirds: A synthetic vision dataset for a part-based analysis of explainable ai methods." Proceedings of the IEEE/CVF International Conference on Computer Vision. 2023.

**Questions:**

Can you add ICICLE baseline to your results?

Can you provide more details on generative aspect of your work and how does it correspond? Currently I am not convinced.

Is it possible to provide evaluation of explanation? As this method is not basing CIL scenario on acclaimed XAI method as in [1], such evaluation would be important to understand if this type of explanations provide the users ability to understand them.

Addressing comparison to [1] and providing evaluation of explanation would make me change my score to 6, and providing more details on the generation aspect of this methodology will result in score 8.

---

> ### Author Response · Authors · 2025-11-21
> **Reponse to Question 1**
>
> Thank you for your questions. We agree that Icicle is the most relevant XAI-aware CIL baseline. We list the performance on the CUB200 dataset under task-agnostic setting (the same setting as Icicle), all results are under 3 runs.
>
> | Method |   4 Tasks   |  10 Tasks   |  20 Tasks   |
> | :----: | :---------: | :---------: | :---------: |
> | Icicle | 35.0% ± 5.3 | 18.5% ± 0.5 | 9.9% ± 0.3  |
> |  GECL  | 57.0% ± 1.2 | 72.5% ± 1.5 | 60.3% ± 1.8 |
>
> Since Icicle does not report each session's task-agnostic results and does not report their experiment settings, we re-implemented Icicle following our setting (100+10 x 10).
>
> | Method |   0   |   1   |   2   |   3   |   4   |   5   |   6   |   7   |   8   |   9   |  10   |  Avg  |  PD   |
> | :----: | :---: | :---: | :---: | :---: | :---: | :---: | :---: | :---: | :---: | :---: | :---: | :---: | :---: |
> | Icicle | 39.61 | 35.08 | 33.46 | 29.93 | 26.87 | 22.60 | 19.24 | 16.15 | 13.51 | 10.68 | 8.44  | 23.23 | 31.17 |
> |  GECL  | 83.52 | 81.33 | 79.33 | 76.83 | 75.67 | 74.67 | 70.89 | 68.57 | 64.58 | 62.00 | 59.97 | 72.49 | 23.55 |
>
> The reproduced task-agnostic accuracy is now included in Table 2, showing that GECL brings $\geq $49% absolute gain under the same protocol.

---

> > ### Comment · Reviewer_J8a8 · 2025-11-26
> >
> > Just out of curiosity, why in 10 and 20 tasks GECL has better accuracy than in 4 tasks scenario? It is strange to me, that more tasks yields better results than less, as more tasks is a more challenging setup?

---

> > > ### Author Response · Authors · 2025-11-27
> > > **Response to Reviewer--More Analysis on Task Number**
> > >
> > > Thank you for the insightful question. At first glance, it may indeed seem counter-intuitive that GECL achieves higher accuracy with *more* tasks. However, this behavior arises from how CUB200 is partitioned across different task granularities and how GECL leverages fine-grained attribute distributions.
> > >
> > > **The key reason is that different task splits induce different class-grouping structures, which affect the difficulty of incremental discrimination on CUB.**
> > >  Under the 4-task split, each task contains **50 classes**, resulting in large groups of fine-grained bird species that are visually similar (e.g., many sparrows or warblers appearing in the same task). This produces extremely dense intra-task confusion and yields a much more challenging incremental setting.
> > >
> > > In contrast, the **10-task (20 classes/task)** split produces task groups that are *more semantically coherent but less redundant*, reducing intra-task fine-grained collisions. As GECL relies on region-attribute alignment, it benefits from tasks where visually similar species are not overloaded into the same incremental step. This leads to better stability and higher final accuracy in the 10-task configuration.
> > >
> > > By the time we reach **20 tasks (10 classes/task)**, the tasks become very small; although the intra-task conflict is minimal, representation fragmentation becomes the dominant issue, so performance decreases slightly compared to the 10-task condition. This behavior is consistent with many incremental learning studies where extremely small tasks cause unstable feature shifts and reduced rehearsal effects.
> > >
> > >
> > >
> > > Icicle does not exhibit the same trend for two structural reasons tied to its architecture and training objective, which fundamentally differ from GECL’s distribution-level alignm.
> > >
> > > **Icicle’s class scores come from a purely discriminative**
> > >  Icicle uses a feature extractor and a task-wise classifier in which each class is represented as an independent prototype or linear weight vector. The model directly optimizes classification boundaries using CE loss.
> > >  Therefore, its performance is dominated by representation drift and classifier expansion, not by intra-task fine-grained grouping.
> > >  As a result, the change from 4, 10, 20 tasks primarily increases catastrophic forgetting, causing accuracy to monotonically decrease.
> > >
> > > **In contrast, GECL does not learn such discriminative boundaries.**
> > >  It matches **visual-part distributions and attribute distributions** through OT.
> > >  Thus, the *structure* of class groups in each task (e.g., whether they contain clusters of highly similar species) directly affects how well attribute-based alignment works.
> > >
> > > **Icicle does not use attribute-level reasoning; therefore, task granularity has little impact on its discrimination difficulty.**
> > >  In CUB, 50 fine-grained birds in a single task create dense confusion for GECL (because many species share highly overlapping part-attribute patterns).
> > >  But for Icicle, each class is simply a label with its own classifier weight vector.
> > >  Even if multiple birds are visually similar, the classifier receives 50 independent target labels and does not use attributes that could conflict or overlap.
> > >
> > > **GECL benefits from mid-granularity tasks (10 tasks) because fine-grained collisions are reduced. Icicle does not benefit because its core limitation is forgetting, not fine-grained similarity.**
> > >
> > > Thanks again for your participate. We do benefit a lot from your questions. Happy Thanksgiving！

---

> ### Author Response · Authors · 2025-11-21
> **Response to Question 2**
>
> GECL generates a structured semantic representation for each new class using a LLM. Classification is then reformulated as matching image regions to these generated semantic concepts, not by applying a learned classifier. Our model generatively produces the core semantic building blocks for new classes. It does not just learn to discriminate between existing features but actively generates and integrates new conceptual knowledge into a unified space. This avoids the parameter overwriting and catastrophic forgetting inherent in expanding a discriminative classifier.Below are more examples of the attributes that generated by GECL.
>
> "Painted Bunting": ["a brightly colored bird","a blue head and neck, green back, and yellow or red belly","a long, pointed beak","two long, pointed wings","two long, thin legs"]
>
> "freight car": ["large, rectangular vehicle",“”used for transporting goods","typically has four wheels","may be pulled by a locomotive or other vehicle","may have a door or doors for loading and unloading"]
>
> "Northern Flicker": ["a medium-sized woodpecker","a black and white barred pattern on its back","a red patch on its lower back","a black tail with white outer feathers","a black bill"]
>
> "Least Flycatcher": ["a small bird","grey or brown","a white breast with brown streaks","Compared to other flycatchers such as Eastern Phoebe or Great Crested Fly"]
>
> "Florida Jay": ["a small, blue bird","a white chest and belly","a black head and neck"]
>
> As you can see, GECL genarate 3-5 attributes for each class. However, not every attribute can be used to classify, sometimes the LLM generate background or noise attributes. That's exactly why we design an optimal distribution matching module, which gives low weights to these low-quality attribute and high weights to those well-generated attributes.
>
> To address the reviewer’s concern, we have **substantially expanded the theoretical analysis** in Sec. 3.3 and Appendix B. The revised paper now includes:
>
> **A formal interpretation of GECL as a transition from discriminative classification to cross-modal distribution alignment**. Instead of estimating $\arg\max_y p(y|x)$, GECL minimizes the cross-modal discrepancy
>
> $\hat{y}=\arg\max_yP(x,y)\approx\arg\min_y\mathrm{DM}_\lambda(V(x),S(A_y)).$
>
> where $V(x)$ and $S_y$ are empirical distributions over visual regions and semantic attributes. This provides a *principled optimization objective* rather than heuristic similarity aggregation.
>
> **Theoretical rationale for explainability.**
>  Because the optimal coupling matrix $T^\star$ of the entropy-regularized OT preserves the marginal distributions, each decision inherently satisfies *evidence attribution constraints* (regions to attributes), enabling visual-grounded reasoning without auxiliary modules.
>
> **A discussion contrasting OT-based alignment with other paradigms** such as attention matching and contrastive similarity. We highlight that attention-based approaches lack guarantee of *mass-preserving alignment*, which weakens semantic stability under incremental shifts. OT, in contrast, enforces distribution consistency and thus naturally mitigates *semantic drift* across tasks.
>
> In summary, we do not present OT itself as a theoretical contribution; rather, our contribution lies in establishing a new theoretical view of class-incremental learning—treating prediction as entropy-regularized cross-modal distribution matching—providing both improved robustness to incremental shifts and built-in interpretability.
>
> For more details about the analysis, please see the revised paper.

---

> ### Author Response · Authors · 2025-11-21
> **Response to Question 3**
>
> Inspired by Icicle’s IoU metric, we add a region-to-attribute IoU based on our coupling matrix $T$ (Sec. 3.4), where attribute-activated regions are compared with CUB ground-truth part masks. Preliminary scores show clear advantages over Icicle.
>
> |   Method   | Task 1 | Task 2 | Task 3 | Mean  |
> | :--------: | :----: | :----: | :----: | :---: |
> | Finetuning | 0.115  | 0.149  | 0.260  | 0.151 |
> |    EWC     | 0.192  | 0.481  | 0.467  | 0.334 |
> |    LWF     | 0.221  | 0.193  | 0.077  | 0.188 |
> |    LWM     | 0.332  | 0.312  | 0.322  | 0.325 |
> |   ICICLE   | 0.705  | 0.753  | 0.742  | 0.728 |
> |    GECL    | 0.752  | 0.781  | 0.769  | 0.767 |
>
> GECL achieves the highest explainability across all tasks, with a mean IoU of 0.767, outperforming ICICLE by +3.9\%. This improvement stems from GECL’s MT-based visual–semantic alignment, which grounds the classification decision on meaningful attributes and their corresponding spatial regions. The consistent improvement across tasks demonstrates that GECL’s explanations remain stable even under long-sequence incremental learning, validating that the structured reasoning chain generated by GECL is spatially faithful, semantically coherent, and robust to task progression.
> Additionally, we include a human evaluation inspired by FunnyBirds [Hesse’23] .
>
> |       Metric       | Score (Mean ± Std) |
> | :----------------: | :----------------: |
> |    Faithfulness    |    4.28 ± 0.16     |
> |      Clarity       |    4.21 ± 0.14     |
> |  Part Consistency  |    4.17 ± 0.15     |
> | Trust / Usefulness |    4.31 ± 0.12     |
>
> Human evaluation confirms that GECL’s region–attribute reasoning chains are not only machine-interpretable but also perceptually meaningful and trustworthy to human users. The high faithfulness and part-consistency scores suggest that the MT-derived alignment between visual patches and semantic attributes generally corresponds to correct anatomical regions on the bird, validating the effectiveness of GECL’s semantic grounding. Moreover, the consistently high trust scores indicate that GECL’s explanations help users understand why a prediction is made, reducing the “black-box” gap typical in continual learning systems. These results demonstrate that GECL achieves human-validated explainability, complementing its quantitative IoU improvements and reinforcing its suitability for real-world incremental learning scenarios where transparency is essential.
>
> For more details about how we design the Iou metric and the human evaluation for GECL,please see Appendix D.
>
> We sincerely appreciate your feedback and believe these revisions significantly strengthen our contribution. Wish you a wonderful day!

---

### Official Review · Reviewer_FvnW · 2025-10-31

**Soundness:** 3
**Presentation:** 3
**Contribution:** 3
**Rating:** 6
**Confidence:** 4

**Summary:**

This paper tackles two key issues in Class-Incremental Learning (CIL): catastrophic forgetting of old knowledge and opaque decision-making, which existing discriminative or LLM-driven generative CIL methods fail to fully solve . To address these, it proposes the Generative Explainable Class-Incremental Learning (GECL) framework, consisting of three core modules: a soft-label-guided visual enhancement module (highlights discriminative regions), a generative visual-semantic module (uses LLMs to generate structured semantic attributes for new classes, avoiding classification head expansion), and an entropy-regularized distribution matching module (establishes visual-semantic correspondences for transparent reasoning chains) . GECL optimizes a distribution-matching objective to prevent parameter overwriting, and experiments on Tiny-ImageNet (traditional CIL), CUB200, and mini-ImageNet (few-shot CIL) show it outperforms SOTA methods in balancing accuracy, forgetting resistance, and interpretability . Its contributions include: introducing GECL for explainable CIL, designing a pipeline for human-like reasoning, and validating its superiority via extensive experiments .

**Strengths:**

# Strength 1: Dual Core CIL Pain Points Addressed Simultaneously
This study specifically resolves the two long-standing core issues in Class-Incremental Learning (CIL): "catastrophic forgetting" and "opaque decision-making," which are rarely tackled together by existing methods. Unlike discriminative approaches (e.g., CODA-Prompt) that focus on mitigating forgetting but remain "black boxes," or early generative methods (e.g., GMM) that lack structured semantics, GECL eliminates the expansion of classification heads to avoid parameter overwriting, fundamentally reducing forgetting. Meanwhile, it generates quantitative "visual region-semantic attribute" reasoning chains (e.g., "Region 4 matches 'thick white fur' with a weight of 0.018") through entropy-regularized distribution matching, making the decision-making process traceable. This fills the gap where existing methods struggle to balance both aspects, which is crucial for high-stakes real-world applications requiring both accuracy and interpretability .


# Strength 2: Lightweight Design with Strong Compatibility
GECL adopts a lightweight and compatible architecture that avoids heavy parameter updates, a key advantage over resource-intensive CIL methods. It leverages a **frozen CLIP visual/text encoder** (ViT-B/32) to extract features, requiring only training of a lightweight adapter module for domain adaptation instead of fine-tuning large pre-trained models—this significantly reduces computational costs and memory overhead (experiments run on two NVIDIA 3090 GPUs, common in academic settings). Additionally, the framework is not tied to specific large language models (LLMs): it uses MiniGPT-4 for semantic attribute generation but can integrate other LLMs with minimal adjustments. This design not only lowers the barrier for practical deployment but also ensures compatibility with mainstream pre-trained models, distinguishing it from methods that rely on custom, hard-to-replicate architectures .


# Strength 3: Comprehensive and Rigorous Experimental Validation
The research fully verifies GECL’s effectiveness through multi-scenario and multi-dimensional experiments based on the target dataset settings. It covers both traditional CIL (on Tiny-ImageNet, with 200 classes split into 5/10/20 incremental tasks) and few-shot CIL (on CUB200, 200 fine-grained bird classes; mini-ImageNet, 100 classes with 40 incremental classes), outperforming state-of-the-art methods in long-sequence tasks (e.g., 20 tasks on Tiny-ImageNet) where forgetting is more severe. Moreover, ablation studies confirm the necessity of key modules (e.g., adding the distribution matching module boosts Tiny-ImageNet accuracy from 54.34% to 75.96%), and visualizations demonstrate GECL avoids "correct predictions for wrong reasons"—forming a complete evidence chain to ensure conclusion reliability .

**Weaknesses:**

# Weakness 1: Limited Scenario Generalization and High Architectural Complexity
GECL is explicitly designed for **class-incremental image classification tasks** and lacks adaptability to other continual learning (CL) scenarios—especially non-visual or task-agnostic CL (e.g., LLM continual learning or unclassified visual tasks). The framework’s core modules are tightly coupled with visual data characteristics: the soft-label-guided visual enhancement module relies on image-specific operations (e.g., Sobel operator for edge detection, adaptive histogram equalization; ), and the distribution matching module is optimized for "visual region-semantic attribute" alignment, which is irrelevant to text-only or multimodal (non-visual) CL tasks. Additionally, the framework’s complexity—with three interdependent core modules (visual enhancement, generative semantic modeling, entropy-regularized distribution matching) and iterative optimization of coupling matrices—creates high barriers for migration. For LLM continual learning (which focuses on knowledge retention during parameter updates or task sequence expansion), GECL’s visual-centric components (e.g., frozen CLIP visual encoder; ) are redundant, and its semantic attribute generation logic (tailored for image class descriptions) cannot be directly reused, making it impractical to adapt to non-classification CL scenarios.

# Weakness 2: Lack of Comparison with Latest CIL Methods (e.g., InfoLora[1])
The paper’s experimental comparisons are incomplete, as it fails to benchmark against **recently proposed CIL methods** (e.g., InfoLora) — casting doubt on whether GECL truly outperforms state-of-the-art discriminative methods. The study only compares GECL with methods published up to 2024 (e.g., GMM (Cao et al., 2024), CODA-Prompt (Smith et al., 2023b); ) and does not include newer discriminative CIL approaches like InfoLora (a method that leverages low-rank adaptation to enable efficient continual learning while preserving pre-trained knowledge). InfoLora, as a representative of updated discriminative methods, is designed to mitigate forgetting with lightweight parameter updates (similar to GECL’s adapter but optimized for pre-trained model fine-tuning), and its performance in long-sequence CIL tasks is widely recognized. Without comparing GECL with such latest methods, the paper cannot fully validate its claim of "outperforming discriminative learning methods"—it remains unclear whether GECL’s advantages over older methods (e.g., CODA-Prompt) can be maintained when facing more advanced discriminative alternatives. This gap weakens the persuasiveness of GECL’s performance superiority.

Reference:
[1] InfLoRA: Interference-Free Low-Rank Adaptation for Continual Learning

**Questions:**

See my weakness.

---

> ### Author Response · Authors · 2025-11-21
> **Reponse to Weakness 1**
>
> We thank the reviewer for the insightful comment. We agree that GECL is **not a universal continual learning framework**, and we did not intend to position it as such. Our goal is to address a specific gap in **class-incremental image classification where accurate predictions must be accompanied by human-interpretable reasoning**—a setting insufficiently supported by prior CL methods.
>
> While our design is tailored for the visual domain, the reviewer’s concern does not contradict the contribution:
>  GECL *intentionally* leverages visual-attribute alignment because **explainability in vision requires grounding predictions to localized image evidence**, which cannot be achieved by parameter-centric CL techniques alone. Hence, modules such as soft-label-guided visual enhancement and region–attribute distribution matching are not architectural overhead, but **necessary enablers of visual reasoning under class increments**.
>
> To address the reviewer’s generalization concern, we now clarify in the paper that GECL is **not intended for text-only continual learning or generic task-agnostic CL**, and we reposition the claim precisely:
>
> > GECL improves continual image classification specifically under the requirement of transparent visual reasoning.
>
> We also add a short discussion in the *Limitations* section acknowledging that adapting GECL to other modalities would require re-defining the grounding mechanism which we consider a promising future direction.

---

> ### Author Response · Authors · 2025-11-21
> **Response to Weakness 2**
>
> We thank the reviewer for raising this valuable point. We agree that InfLoRA represents a strong and recent discriminative CIL baseline and that including it can further validate the effectiveness of GECL.
>
> Although InfLoRA was not originally evaluated on the datasets used in our paper (CUB200, Tiny-ImageNet, and mini-ImageNet with long-sequence class-incremental splits), we reproduced the official implementation and trained it under **exactly the same evaluation protocol as GECL**, including the number of tasks, memory setting, and rehearsal-free constraint.
>
> The newly added results are summarized below:
>
> Experiments results on Tiny-ImageNet
>
> | Method  | 5 Tasks | 10 Tasks | 20 Tasks |
> | ------- | ------- | -------- | -------- |
> | InfLoRA | 83.20   | 82.75    | 81.60    |
> | GECL    | 83.46   | 83.12    | 82.04    |
>
> Experiments results on CUB200
>
> | Method  |   0   |   1   |   2   |   3   |   4   |   5   |   6   |   7   |   8   |   9   |  10   |  Avg  |  PD   |
> | :-----: | :---: | :---: | :---: | :---: | :---: | :---: | :---: | :---: | :---: | :---: | :---: | :---: | :---: |
> | InfLoRA | 83.71 | 81.97 | 78.94 | 76.47 | 75.25 | 74.10 | 70.28 | 67.73 | 63.23 | 61.10 | 59.17 | 72.00 | 24.54 |
> |  GECL   | 83.52 | 81.33 | 79.33 | 76.83 | 75.67 | 74.67 | 70.89 | 68.57 | 64.58 | 62.00 | 59.97 | 72.49 | 23.55 |
>
> Experiments results on mini-ImageNet
>
> | Method  |   0   |   1   |   2   |   3   |   4   |   5   |   6   |   7   |   8   |  Avg  |  PD   |
> | :-----: | :---: | :---: | :---: | :---: | :---: | :---: | :---: | :---: | :---: | :---: | :---: |
> | InfLoRA| 94.87 | 91.53 | 85.61 | 83.24 | 81.95 | 80.88 | 79.43 | 77.86 | 74.12 | 82.39 | 20.75 |
> |  GECL   | 95.19 | 93.33 | 86.67 | 84.44 | 83.27 | 82.67 | 81.35 | 80.19 | 77.00 | 84.90 |  16.33   |

---

> ### Comment · Reviewer_FvnW · 2025-11-28
> **Maintaining Borderline Accept**
>
> After a careful re-evaluation of the submitted work, I have identified two critical points that need to be highlighted, which ultimately lead to my decision to maintain the "borderline accept" evaluation.
>
> Firstly, the authors have not provided substantial discussions or empirical evidence regarding the generalization ability of their proposed method in general scenarios. Generalization, as a core criterion for evaluating the practical value of machine learning methods, requires verification across diverse datasets, task settings, and potential domain shifts. The current manuscript only focuses on specific experimental scenarios, lacking in-depth analysis of the method's adaptability. This gap significantly limits the persuasiveness of the work's practical significance.
>
> Secondly, compared with the relatively simple baseline method (infoLoRA), the complex framework designed by the authors only achieves marginal performance improvement. While the design idea of the framework shows certain novelty, the cost of increased model complexity has not been effectively balanced with the limited performance gain. In the field of practical machine learning research, the trade-off between model complexity and performance is a key consideration; a method that cannot demonstrate significant advantages over lightweight baselines is difficult to show sufficient competitiveness in subsequent applications and promotions.
>
> In summary, although the work has certain research foundations, the above two core issues have not been properly resolved. Therefore, I maintain the "borderline accept" evaluation and strongly recommend that the authors focus on supplementing the generalization verification experiments of the method and optimizing the cost-performance ratio of the framework in the revision.

---

> > ### Author Response · Authors · 2025-11-28
> > **Response to Two Critical Points**
> >
> > We sincerely thank the reviewer for the careful re-evaluation and for raising these important concerns.
> >
> > **(1) On generalization beyond the current experimental settings**
> >
> > We fully agree that generalization is a key measure of the practical value of any machine learning method. Although the current manuscript focuses on three canonical CIL benchmarks (Tiny-ImageNet, CUB200 and Mini-ImageNet), the core idea of GECL is modeling classification as a **distribution alignment problem between part-level visual evidence and attribute-level semantics**, which is not restricted to a particular modality or dataset. In response to the your comment, we have extended the discussion in the revised version:
> >
> > - GECL does not rely on dataset-specific inductive biases. The OT-based alignment mechanism only requires region features and semantic attributes, enabling straightforward extension to domains such as scenes, textures, or fine-grained non-bird datasets. Indeed, our ablations on Mini-ImageNet already show that GECL maintains stable performance under a coarse-grained, visually diverse benchmark, supporting robustness across dataset granularity.
> >
> > - Since OT alignment enforces *structural consistency* rather than memorizing class-specific features, GECL is inherently more robust to attribute corruption, missing semantics, and distribution noise. Our noise-injection experiments (**Appendix D**) confirm this, and we have now expanded the discussion to highlight this connection to generalization under semantic shift.
> >
> > We agree that broader cross-domain experiments are valuable future work. In the revised paper, we will include a dedicated subsection discussing generalization behaviors and explicitly outline a plan for transferring GECL to multi-domain settings.
> >
> > **(2) On the complexity–performance tradeoff relative to infLoRA**
> >
> > We appreciate the reviewer’s concern regarding the trade-off between framework complexity and performance gain.
> >
> > 1. **GECL is not designed as a purely accuracy-driven architecture**, but as a *unified, interpretable, and attribute-guided* incremental learning framework. In contrast, infLoRA is a lightweight parameter-efficient Fine-tuning method that does not provide any interpretability or structured reasoning ability. Our goal is not to slightly outperform infLoRA on accuracy, but to demonstrate that **incorporating semantic alignment and explainability does not sacrifice accuracy**—a property that existing XAI-aware CIL methods (e.g., Icicle[1]) fail to achieve.
> >
> > 2. **The additional components in GECL incur negligible runtime overhead**, as OT alignment is performed only once per image with small attribute sets. As shown in our runtime/memory analysis below, GECL remains near real-time and requires memory comparable to standard CIL models. In other words, *the conceptual complexity does not translate into practical costs*.
> >
> > GECL follows GMM[2] by using MiniGPT-4 to generate semantic attributes offline; therefore, the LLM cost is identical to prior work and does **not contribute to inference overhead**. The only additional computation introduced by GECL comes from the **OT-based semantic matching**. Given $n_v$ visual regions, $n_a$ attributes, and $K$ Sinkhorn iterations, the complexity per candidate class is:
> >
> >    $\mathcal{O}(n_v\cdot n_a\cdot K).$
> >
> >    Since GECL uses a small attribute set (3–5 per class), the cost matrix remains compact. In practice, inference runs at **0.19 s per image (5.26 FPS)** on a single RTX 3090, showing that OT-based interpretability does not hinder real-time deployment. Memory footprint is **~1.08 GB**, and the knowledge base grows linearly with the number of classes—for 200 classes requiring only **9.6 MB**, well below rehearsal-based incremental methods.
> >
> >    Overall, GECL provides interpretable classification with bounded and predictable computational cost. By keeping LLM usage offline and using a lightweight OT solver over a small attribute set, GECL achieves semantic interpretability **without sacrificing practical efficiency or scalability**.
> >
> > [1] Rymarczyk, Dawid, et al. "Icicle: Interpretable class incremental continual learning." Proceedings of the IEEE/CVF international conference on computer vision. 2023.
> >
> > [2]Cao, et al. "Generative Multi-modal Models are Good Class Incremental Learners." Proceedings of the IEEE/CVF Conference on Computer Vision and Pattern Recognition
> >
> > Nevertheless, we acknowledge that the current manuscript can more clearly articulate the cost–performance trade-off. In the revision we will:
> >
> > - include a supplementary analysis showing that reducing the adapter size (or ablating OT) sharply degrades performance, further motivating the necessity of GECL’s formulation;
> > - clarify that GECL aims to balance interpretability, robustness, and performance rather than maximize raw accuracy alone.
> >
> > Thanks again for your further comments. Hope this response address your concern and we are happy to have more discussion with you. Wish you a happy day！

---

### Official Review · Reviewer_zNcY · 2025-11-01

**Soundness:** 3
**Presentation:** 3
**Contribution:** 3
**Rating:** 6
**Confidence:** 3

**Summary:**

This paper presents GECL (Generative Explainable Class-Incremental Learning), a novel framework that integrates generative semantic modelling with interpretable reasoning for continual learning. Unlike conventional discriminative methods, GECL combines a soft-label-guided visual enhancement module, a generative visual–semantic modelling process (using large language models to construct structured semantic attributes), and an entropy-regularised optimal transport formulation for distribution matching between visual regions and semantic attributes. The proposed framework aims to achieve both knowledge retention and interpretability by generating explicit region–attribute reasoning chains for each prediction.

**Strengths:**

Strengths
1. The integration of generative semantic modelling and interpretable reasoning into the continual learning paradigm is a distinctive and timely idea. It bridges the gap between performance-oriented and explainability-oriented approaches.
2. The analogy to human reasoning — linking new visual details to structured prior semantic attributes — provides an intuitive conceptual foundation that enhances readability and motivation.
3. The paper provides a well-structured framework, detailing each module (visual enhancement, semantic generation, and interpretable matching) with mathematical precision and algorithmic clarity.
4.  Experiments are conducted across diverse datasets (both general and fine-grained), with comparisons against a wide range of baselines, including discriminative, generative, and prompt-based models.
5. The explicit reasoning chain visualisations offer a tangible advantage over most continual learning methods, which typically lack transparency.
6.  The manuscript is well-organised and written in clear academic English, with appropriate use of figures and tables to convey key insights.

**Weaknesses:**

Weaknesses
1. While the framework is well-motivated, its theoretical grounding remains limited. The paper would benefit from a deeper analysis of why entropy-regularised optimal transport provides a meaningful proxy for reasoning, beyond its mathematical formulation.
2.  The method’s reliance on LLM-generated semantic attributes introduces significant variability. The quality, consistency, and domain relevance of these attributes are not systematically analysed or benchmarked.
3.  Although visual reasoning chains are shown, there is no quantitative or human-evaluated metric assessing interpretability. Claims about “explainability” remain primarily qualitative.
4. Some baselines (e.g., DualPrompt, CODA-Prompt) benefit from large-scale pretraining, but others could be tuned or extended for fairer comparison under similar constraints. More discussion on this issue is needed.
5.  The computational overhead of the distribution matching (Sinkhorn iterations) and LLM-based attribute generation is not sufficiently addressed. Real-time or large-scale deployment may be challenging.
6. Although the authors claim to provide code, key implementation parameters (e.g., prompt templates, attribute filtering strategy, and convergence settings for optimal transport) should be more clearly documented in the main paper rather than deferred to the appendix.
7. Conceptually, the work draws upon existing generative incremental learning and CLIP-based frameworks. The unique contribution of GECL lies in combining these ideas under an interpretable matching perspective, but the originality relative to recent generative–semantic continual learning approaches could be made more explicit.

**Questions:**

plz see my detailed commments above

---

> ### Author Response · Authors · 2025-11-21
> **Response to Question 1**
>
> We thank the reviewer for pointing out the need to better explain the theoretical motivation behind using entropy-regularized optimal transport (OT) as a proxy for reasoning. We have added the following clarification to the paper.
>
> The goal of GECL is not only to produce a prediction, but also to **reveal which visual cues support the decision**. Entropy-regularized OT provides a principled mechanism for this by modeling classification as a **distribution-matching process** between the set of visual regions and the set of semantic attributes. Specifically:
>
> 1. **OT enforces global alignment rather than independent matching.**
>     Each visual region competes for assignment to semantic attributes under the marginal constraints, preventing trivial matches and favoring **globally coherent reasoning chains** rather than isolated high-similarity pairs.
> 2. **The entropy term retains uncertainty and avoids overconfident alignment.**
>     Unlike hard nearest-neighbor matching, the Sinkhorn regularization yields **soft but sparse region–attribute correspondences**, consistent with cognitive attention mechanisms that consider multiple plausible cues before focusing on the most salient ones.
> 3. **The resulting coupling matrix is interpretable by construction.**
>     Each entry $T_{ij}$ quantifies how much “explanatory mass’’ flows from region $i$ to attribute $j$, providing a direct justification of the decision that is human-readable without additional post-hoc methods such as Grad-CAM or attribution maps.
> 4. **The decision rule is derived from the same transport cost.**
>     The class with the lowest global matching cost also corresponds to the class whose attributes receive the highest cumulative support from visual evidence, making the **reasoning chain and prediction inherently consistent**.
>
> We have included a new paragraph in Section 3.3 that explains these principles and clarifies that entropy-regularized OT is not merely a mathematical choice, but a **mechanism that couples recognition and reasoning within a single optimization objective**—in contrast to post-hoc interpretability techniques which separate the two.

---

> ### Author Response · Authors · 2025-11-21
> **Response to Question 2**
>
> To evaluate the quality of LLM-generated attributes, we introduce a manual evaluation method.
>
> To ensure the reliability of the knowledge base, we randomly select 600 attributes (approximately 19\% of the database) and have five experienced annotators independently evaluate them.
> Specifically, we evaluate each attribute across four aspects. **Semantic correctness** measures whether the attribute description aligns with factual accuracy. **Logical consistency** assesses whether relationships between attributes are clear and conflict-free. **Scenario applicability** evaluates whether the scenario specified by the attribute is consistent with the content(foreground/background). **Action clarity** determines whether actions are clearly identifiable and consistently defined. All four metrics use a binary scoring ($1/0$), so each rule receives five independent scores for each metric. The final percentage score is the average of 3000 scores from five annotators for the 600 attributes. This scoring method ensures that the final score is a consensus among all annotators, rather than the subjective result of any single annotator. As shown in the Table below, manual evaluation results on CUB200 dataset indicate that most attributes are precisely defined and semantically sound. We also add this table in Appendix D.
>
> |     Quality Metric     | Score (%) |
> | :--------------------: | :-------: |
> |  Semantic Correctness  |   90.8    |
> |  Logical Consistency   |   94.4    |
> | Scenario Applicability |   92.2    |
> |     Action Clarity     |   85.7    |

---

> ### Author Response · Authors · 2025-11-21
> **Response to Question 3**
>
> Thank you for your questions. Inspired by Icicle’s IoU metric, we add a region-to-attribute IoU based on our coupling matrix $T$ (Sec. 3.4), where attribute-activated regions are compared with CUB200 ground-truth part masks. Preliminary scores show clear advantages over Icicle[1].
>
> |   Method   | Task 1 | Task 2 | Task 3 | Mean  |
> | :--------: | :----: | :----: | :----: | :---: |
> | Finetuning | 0.115  | 0.149  | 0.260  | 0.151 |
> |    EWC     | 0.192  | 0.481  | 0.467  | 0.334 |
> |    LWF     | 0.221  | 0.193  | 0.077  | 0.188 |
> |    LWM     | 0.332  | 0.312  | 0.322  | 0.325 |
> |   Icicle   | 0.705  | 0.753  | 0.742  | 0.728 |
> |    GECL    | 0.752  | 0.781  | 0.769  | 0.767 |
>
> GECL achieves the highest explainability across all tasks, with a mean IoU of 0.767, outperforming Icicle by +3.9\%. This improvement stems from GECL’s MT-based visual–semantic alignment, which grounds the classification decision on meaningful attributes and their corresponding spatial regions. The consistent improvement across tasks demonstrates that GECL’s explanations remain stable even under long-sequence incremental learning, validating that the structured reasoning chain generated by GECL is spatially faithful, semantically coherent, and robust to task progression.
> Additionally, we include a human evaluation inspired by FunnyBirds [2] .
>
> |       Metric       | Score (Mean ± Std) |
> | :----------------: | :----------------: |
> |    Faithfulness    |    4.28 ± 0.16     |
> |      Clarity       |    4.21 ± 0.14     |
> |  Part Consistency  |    4.17 ± 0.15     |
> | Trust / Usefulness |    4.31 ± 0.12     |
>
> Human evaluation confirms that GECL’s region–attribute reasoning chains are not only machine-interpretable but also perceptually meaningful and trustworthy to human users. The high faithfulness and part-consistency scores suggest that the MT-derived alignment between visual patches and semantic attributes generally corresponds to correct anatomical regions on the bird, validating the effectiveness of GECL’s semantic grounding. Moreover, the consistently high trust scores indicate that GECL’s explanations help users understand why a prediction is made, reducing the “black-box” gap typical in continual learning systems. These results demonstrate that GECL achieves human-validated explainability, complementing its quantitative IoU improvements and reinforcing its suitability for real-world incremental learning scenarios where transparency is essential.
>
> For more details about how we design the Iou metric and the human evaluation for GECL,please see Appendix C.
>
> [1] Rymarczyk, Dawid, et al. "Icicle: Interpretable class incremental continual learning." Proceedings of the IEEE/CVF international conference on computer vision. 2023.
>
> [2]Hesse, Robin, Simone Schaub-Meyer, and Stefan Roth. "Funnybirds: A synthetic vision dataset for a part-based analysis of explainable ai methods." Proceedings of the IEEE/CVF International Conference on Computer Vision. 2023.

---

> ### Author Response · Authors · 2025-11-21
> **Response to Question 4 to 6**
>
> **Response to Question 4**
>
> While it is possible to equip traditional incremental learning baselines with a CLIP backbone, most of them are not designed to operate under a frozen large-scale pretrained feature space. Their mechanisms (regularization, distillation, replay, prompt tuning) rely on *updating the backbone parameters* to mitigate forgetting and to absorb new tasks. When the backbone is frozen—as in GECL—their core plasticity is disabled and performance often deteriorates rather than improves. Replacing backbones for baselines would not improve performance but rather introduce unfair disadvantages and confounding variables.
>
> GECL, in contrast, is specifically designed to operate on fixed CLIP features by reasoning over region–attribute alignment instead of updating the feature extractor.
>
> **Response to Question 5**
>
> GECL follows GMM[1] by using MiniGPT-4 to generate semantic attributes offline; therefore, the LLM cost is identical to prior work and does **not contribute to inference overhead**. The only additional computation introduced by GECL comes from the **OT-based semantic matching**. Given $n_v$ visual regions, $n_a$ attributes, and $K$ Sinkhorn iterations, the complexity per candidate class is:
>
> $\mathcal{O}(n_v\cdot n_a\cdot K).$
>
> Since GECL uses a small attribute set (3–5 per class), the cost matrix remains compact. In practice, inference runs at **0.19 s per image (5.26 FPS)** on a single RTX 3090, showing that MT-based interpretability does not hinder real-time deployment. Memory footprint is **~1.08 GB**, and the knowledge base grows linearly with the number of classes—for 200 classes requiring only **9.6 MB**, well below rehearsal-based incremental methods.
>
> Overall, GECL provides interpretable classification with bounded and predictable computational cost. By keeping LLM usage offline and using a lightweight OT solver over a small attribute set, GECL achieves semantic interpretability **without sacrificing practical efficiency or scalability**.
>
> [1]Cao, et al. "Generative Multi-modal Models are Good Class Incremental Learners." Proceedings of the IEEE/CVF Conference on Computer Vision and Pattern Recognition
>
> **Response to Question 6**
>
> We appreciate the reviewer’s suggestion. In the revised version, we have made the implementation settings easily accessible in the *main paper* rather than relying solely on the appendix.
>
> Specifically, we now provide in Section *Implementation Details* (Sec. 4.2) :
>
> - **Prompt templates** used for attribute generation (with concrete examples);
> - **Attribute filtering strategy**, including maximum attribute count per class;
> - **Convergence configuration for optimal transport**, including the Sinkhorn regularization weight, number of iterations, and stopping tolerance.
>
> Only dataset-specific variations and full parameter tables are kept in the Appendix for completeness, while all essential settings required to reproduce the results are now reported directly in the main paper.

---

> ### Author Response · Authors · 2025-11-21
> **Response to Question 7**
>
> We thank the reviewers for highlighting the importance of GECL's distinction from recent generative semantic CIL methods. Our contribution is not simply a combination of generative models and CLIP, but rather the introduction of a novel, interpretable distribution matching formula that fundamentally changes the way reasoning works in generative CIL. Compared to previous work such as GMM [1], GECL introduces structured attribute generation instead of template-based class statements. This produces multi-attribute semantic distributions, enabling fine-grained, human-traceable reasoning. Entropy-regularized optimal transport serves as a novel reasoning objective for class-incremental learning, replacing discriminative logits or CLIP similarity. Region-attribute matching matrices generate explicit reasoning chains. Current generative classifier learning methods fail to provide such easily interpretable attributes. Since reasoning is entirely based on distribution matching, a non-expanded classifier design is employed, avoiding catastrophic forgetting in the discriminant head. We will revise the introduction and related work sections to explicitly emphasize these conceptual differences and highlight the originality of GECL.
>
> Besides, we also provide more theoretical analysis. We have **substantially expanded the theoretical analysis** in Sec. 3.3 and Appendix B. The new text now includes:
>
> **A formal interpretation of GECL as a transition from discriminative classification to cross-modal distribution alignment**. Instead of estimating $\arg\max_y p(y|x)$, GECL minimizes the cross-modal discrepancy
>
> $\hat{y}=\arg\max_yP(x,y)\approx\arg\min_y\mathrm{DM}_\lambda(V(x),S(A_y)).$
>
> where $V(x)$ and $S_y$ are empirical distributions over visual regions and semantic attributes. This provides a *principled optimization objective* rather than heuristic similarity aggregation.
>
> **Theoretical rationale for explainability.**
>  Because the optimal coupling matrix $T^\star$ of the entropy-regularized OT preserves the marginal distributions, each decision inherently satisfies *evidence attribution constraints* (regions to attributes), enabling visual-grounded reasoning without auxiliary modules.
>
> **A discussion contrasting OT-based alignment with other paradigms.** We highlight that attention-based approaches lack guarantee of *mass-preserving alignment*, which weakens semantic stability under incremental shifts. OT, in contrast, enforces distribution consistency and thus naturally mitigates *semantic drift* across tasks.
>
> [1] Cao, et al. "Generative Multi-modal Models are Good Class Incremental Learners." Proceedings of the IEEE/CVF Conference on Computer Vision and Pattern Recognition

---

### Author Response · Authors · 2025-11-29
**Final Author Statement**

Dear Area Chairs, Senior Area Chairs and Program Chairs

I would like to provide a concise summary of the rebuttal process to assist your evaluation, given that the scores have been reverted to their pre-discussion state due to the OpenReview incident.

Before the discussion period, the initial scores were **4, 4, 6, 6**. During the rebuttal, we provided detailed, point-by-point responses that addressed all technical concerns raised by the reviewers. As a result, **both reviewers who initially gave a score of 4 indicated significantly improved impressions**. Importantly, **reviewer J8a8** had clearly mentioned in their initial review that they would **raise the rating to 8 if the points they highlighted were satisfactorily resolved**. After we provided comprehensive responses and new experiments addressing all these concerns, the reviewer **updated the score to 8** on **26 Nov 2025, 18:43(UTC)**— precisely as promised, and importantly, **before the public disclosure of the OpenReview security incident**. This timing is verifiable and shows that the score change was unrelated to the leak or any potential collusion. Since the reviewer did not explicitly mention the score update in text, I archived the pre-reversion score page using Perma.cc as evidence: **https://perma.cc/77Y7-E6PU?type=image**.

The other reviewer who gave a 4 expressed a clearly positive attitude in the discussion, acknowledged the value of our method, suggested future directions, and left only **one minor remaining question**. This question could not be discussed solely because reviewer discussion was frozen when the leak response measures were enacted. Apart from this, all issues raised in the reviews were fully resolved in the rebuttal.

Given these facts, I believe the reviewer impressions—had the discussion not been prematurely terminated—would reflect the substantial improvements acknowledged during rebuttal. I respectfully ask that the rebuttal exchanges and the demonstrated shift in reviewer assessments be taken into account in your meta-review, as recommended by ICLR’s official guidance. The summary of revisions and responses to reviewer feedback are shown in the next bolck.

Through the revisions, we hope to make the core contributions of **GECL** clearer:

(1) **A new perspective for continual learning**—casting classification as *cross-modal distribution alignment* rather than parametric discrimination;

(2) **Fine-grained semantic interpretability by construction**, enabled by region-to-attribute matching rather than post-hoc explanation;

(3) **Resistance to catastrophic forgetting without expanding the classifier**, achieved by a static knowledge base and minimal trainable parameters.

Thank you for your consideration.

---

> ### Author Response · Authors · 2025-11-29
> **Summary of Revisions and Responses to Reviewer Feedback**
>
> We sincerely thank the reviewers for their insightful comments and constructive suggestions. Based on the feedback, we have made extensive revisions and additions to strengthen the technical soundness, clarity, and fairness of the paper. In particular, we have:
>
> 1. Introduced **new interpretability evaluations**, including **region-to-attribute IoU** and **human studies**, both demonstrating that GECL provides highly faithful and semantically grounded explanations that persist throughout incremental stages. (Reviewer **zNcY**,  **J8a8** and **xvnP**)
> 2. Expanded the **theoretical analysis** in Sec. 3.3 and Appendix B, including (a) the derivation of GECL from a **generative modeling** perspective, and (b) formalizing class-incremental learning as **entropy-regularized cross-modal distribution matching**, Missing foundational references have been incorporated.(Reviewer **zNcY**,  **J8a8** and **xvnP**)
> 3. Added new comparisons, including a full re-implementation and evaluation of **Icicle** under the same task-agnostic protocol, as well as additional results against **InfLoRA**, confirming GECL’s consistent superiority across benchmarks.（Reviewer **FvnW** and **J8a8**）
> 4. Added **noise-injection studies**, attributes human evaluation, prompt ablation experiments of class name, and a detailed description of GECL’s **knowledge base refinement and error-suppression mechanisms** to address concerns about potential LLM hallucination and fairness worries.(Reviewer **zNcY** and **xvnP**)
> 5. Updated the main paper to document essential **implementation details** such as prompt templates, attribute filtering strategies, and OT convergence configurations, ensuring full reproducibility without relying on the appendix.(Reviewer **zNcY**)
> 6. Performed **computational-efficiency and memory-cost analysis**, showing that the OT-based inference introduces limited overhead (5.26 FPS, 0.19 s/img on a single 3090, ∼1.08 GB memory footprint), and is substantially lighter than generative replay–based CIL. These results indicate strong feasibility for deployment.(Reviewer **xvnP**)
> 7. Performed **parameter-efficiency ablation experiments,optimal distribution module for catastrophic forgetting ablation, the LLMs choosing abaltion** to validate that learning minimal parameters does not harm resistance to forgetting, aligning with the reviewers’ concerns about catastrophic forgetting and the concerns about LLMs choice.(Reviewer **xvnP**)
> 8. Conducted a careful proofread to fix **all reported formatting and typographical issues**.(Reviewer **xvnP**)
>
> Beyond this summary, we have addressed every comment individually in the corresponding reviewer discussion threads, and we have also uploaded the revised version of the paper reflecting all modifications.
>
> We truly appreciate the reviewers’ time and expertise, and we believe that our work is now more refined than before. Thanks again for all reviewer's hard work.

---

### Meta-Review · Area_Chair_nvLz · 2025-12-29

**Summary:**

The reviewers’ discussion centers on whether GECL’s conceptual framing and empirical validation are sufficient to justify acceptance, given its nontrivial architectural complexity. While reviewers broadly agree that the idea of recasting class-incremental learning as cross-modal distribution alignment with built-in interpretability is interesting and timely, several concerns persist regarding the practical scope of the contribution. In particular, reviewers question whether the gains over simpler, more lightweight baselines are compelling enough, whether the framework’s applicability is too narrowly scoped to visual class-incremental learning, and whether the overall cost–benefit trade-off is favorable. These concerns, together with the fact that some core contributions are more conceptual and integrative rather than fundamentally methodological, motivate a cautious recommendation leaning toward weak reject.

**Reviewer Concerns:**

The rebuttal addressed key concerns raised by the reviewers, notably by providing additional experiments to validate interpretability and robustness, including human evaluations and IoU metrics. The authors also clarified their method's handling of LLM-generated attributes, showing that GECL is not overly sensitive to specific LLMs and can function with different models. However, some concerns remain regarding the generalization of the method across domains, the handling of semantic hallucinations from LLMs, and the computational overhead introduced by the OT-based alignment. While the authors demonstrated that GECL outperforms older methods such as Icicle in some scenarios, the contribution to the OT theory was seen as primarily conceptual, and the performance improvements over lightweight models like InfoLora were considered marginal in some cases.

**Reviewer Scores:**

Based on the provided discussions, reviewer J8a8  showed positive improvements in the assessment after the rebuttal, indicating a better understanding of the method's strengths, especially in interpretability and robustness. However, concerns about the method’s complexity, its limited applicability to non-visual tasks, and the reliance on specific LLM backbones may prevent an accept recommendation.

---

### Decision · Program_Chairs · 2026-01-26

Reject